# PET-CT in Clinical Adult Oncology—V. Head and Neck and Neuro Oncology

**DOI:** 10.3390/cancers14112726

**Published:** 2022-05-31

**Authors:** Richard H. Wiggins, John M. Hoffman, Gabriel C. Fine, Matthew F. Covington, Ahmed Ebada Salem, Bhasker R. Koppula, Kathryn A. Morton

**Affiliations:** 1Department of Radiology and Imaging Sciences, University of Utah, Salt Lake City, UT 84132, USA; richard.wiggins@hsc.utah.edu (R.H.W.); john.hoffman@hci.utah.edu (J.M.H.); gabriel.fine@hsc.utah.edu (G.C.F.); matthew.covington@hsc.utah.edu (M.F.C.); ahmed.salem@utah.edu (A.E.S.); bhasker.koppula@hsc.utah.edu (B.R.K.); 2Department of Radiodiagnosis and Intervention, Faculty of Medicine, Alexandria University, Alexandria 21526, Egypt; 3Intermountain Healthcare Hospitals, Summit Physician Specialists, Murray, UT 84123, USA

**Keywords:** PET, FDG, head and neck cancer, brain tumors, squamous cell carcinoma, cervical esophageal cancer, salivary tumors, encephalitis, leptomeningeal carcinomatosis, CNS lymphoma, meningioma

## Abstract

**Simple Summary:**

Positron emission tomography (PET), typically combined with computed tomography (CT) has become a critical advanced imaging technique in oncology. With PET-CT, a radioactive molecule (radiotracer) is injected in the bloodstream and localizes to sites of tumor because of specific cellular features of the tumor that accumulate the targeting radiotracer. The CT scan, performed at the same time, provides information to facilitate attenuation correction, so that radioactivity from deep or dense structures can be better visualized, but with head and neck malignancies it is critical to provide correlating detailed anatomic imaging. PET-CT has a variety of applications in oncology, including staging, therapeutic response assessment, restaging, and surveillance. This series of six review articles provides an overview of the value, applications, and imaging and interpretive strategies of PET-CT in the more common adult malignancies. The fifth report in this series provides a review of PET-CT imaging in head and neck and neuro oncology.

**Abstract:**

PET-CT is an advanced imaging modality with many oncologic applications, including staging, assessment of response to therapy, restaging, and longitudinal surveillance for recurrence. The goal of this series of six review articles is to provide practical information to providers and imaging professionals regarding the best use of PET-CT for specific oncologic indications, and the potential pitfalls and nuances that characterize these applications. In addition, key tumor-specific clinical information and representative PET-CT images are provided to outline the role that PET-CT plays in the management of oncology patients. Hundreds of different types of tumors exist, both pediatric and adult. A discussion of the role of FDG PET for all of these is beyond the scope of this review. Rather, this series of articles focuses on the most common adult malignancies that may be encountered in clinical practice. It also focuses on FDA-approved and clinically available radiopharmaceuticals, rather than research tracers or those requiring a local cyclotron. The fifth review article in this series focuses on PET-CT imaging in head and neck tumors, as well as brain tumors. Common normal variants, key anatomic features, and benign mimics of these tumors are reviewed. The goal of this review article is to provide the imaging professional with guidance in the interpretation of PET-CT for the more common head and neck malignancies and neuro oncology, and to inform the referring providers so that they can have realistic expectations of the value and limitations of PET-CT for the specific type of tumor being addressed.

## 1. Introduction

PET-CT is an invaluable advanced diagnostic imaging modality in oncology with a variety of applications, including initial staging of cancer, assessment of response to therapy, restaging and longitudinal surveillance for recurrence. In addition, the modality is useful in identifying the primary tumor and extent of disease in patients who present with a metastatic lesion from a carcinoma of unknown primary, or in those with paraneoplastic manifestations. Historically, clinical oncologic PET utilized ^18^F fluorodeoxyglucose (FDG) almost exclusively. However, recent years have seen the advent of new PET radiopharmaceuticals for specific oncologic applications that are now FDA approved and clinically available. These include NETSPOT^®^ (^68^Ga DOTATATE, AAA Novartis, NY, USA) and Detectnet^®^ (^64^Cu DOTATATE, Curium US, LLC, Maryland Heights, MO) for well-differentiated neuroendocrine tumor, as well as Axumin^®^ (^18^F anti-1-amino-3-^18^F-fluorocyclobutane-1-carboxylic acid, also knowns as FACBC, Blue Earth Diagnostics, Oxford, UK) and the recently FDA approved PYLARIFY^®^ (^18^F piflufolastat, also called DCFPyL or PyL, Lantheus, Billerica, MA, USA) for prostate cancer imaging. These radiopharmaceuticals, combined with widespread availability, technological improvements in instrumentation such as digital PET-CT, and acquired global experience have placed PET-CT at the forefront of oncologic imaging. Although these agents are all FDA approved for specific indications, there is potential expanded use of these agents for other types of malignancies, particularly with respect to brain tumors.

This review article is the fifth in a series of six articles addressing PET-CT in clinical adult oncology. The focus of this specific report is on the use of PET-CT for tumors of the head and neck, as well as of the brain. The goal of this article is to provide practical information to referring providers, radiologists, nuclear medicine practitioners and their trainees regarding the best use of PET-CT for these indications, and the potential pitfalls and nuances that characterize these applications. This article also focuses on FDA-approved and clinically accessible radiopharmaceuticals, rather than research tracers or those that require a local cyclotron. In addition, it is acknowledged that there is a large body of literature relating PET characteristics to prognostic factors with specific cancers. This is also not emphasized, as it arguably rarely impacts clinical decision making and patient care algorithms. Because PET-MR scanners are in limited use in the U.S., the focus herein is on PET-CT because if its widespread availability. However, basic principles described are also applicable to PET-MR, which has significant advantages in head and neck and the neuro oncology imaging. The targeted readers for this review are imaging providers, including radiologists, nuclear medicine physicians, and their trainees. The information is also important to medical and surgical care professionals caring for or treating adult cancer patients.

## 2. Head and Neck Cancer

Cutaneous tumors of the head and neck will be addressed separately in the sixth review article of this series.

### 2.1. Technical Notes, Nuances and Overview

FDG PET-CT plays an important role in the initial staging, assessment of response to treatment, and surveillance and management of recurrent disease in head and neck cancer (HNCa) [1]. In addition to a standard whole-body protocol for assessment of distant metastases, the addition of a thin-section, high-resolution, contrast-enhanced acquisition has been shown to be of additional value in improving lesion definition and detection of potential nodal disease [2,3,4]. PET-CT is less affected by dental amalgam, offering a significant advantage in this respect over CT and MRI. The PET-CT scanning technique may differ slightly, depending on the manufacturer’s recommendations but can also be modified by the users for improved performance. For example, for head and neck imaging, one can apply a 256 × 256 matrix acquired at 1.5 zoom with longer imaging times to compensate for finer spatial sampling [3]. Head positioning and lack of motion between the PET and CT is critically important. 

FDG PET-CT is also being increasingly used for radiation treatment planning in patients with HNCa although standardized procedures and the impact on therapeutic approach and outcome have not been clearly established [5,6]. For this application, patients may be scanned on a flatbed (duplicating radiation treatment conditions), with a thermoplastic treatment mask and an oral appliance in place, with positioning by the radiation treatment dosimetrist. The acquired images can be imported into the modality workstation for dose planning. 

Initial staging of HNCa with FDG PET-CT requires accurate identification of involved lymph nodes. Reactive lymph nodes may be metabolically active and false positive lymph nodes may occur in response to upper respiratory illness, inflammatory lesions on the skin or in the mouth or throat, sinus infection, and/or periodontal disease (Figure 1). Normal patients, particularly those who are younger, often have symmetrical mild uptake in high jugulodigastric lymph nodes. These conditions may contribute to false positive scans. HNCa primary tumors may ulcerate and/or lead to surrounding inflammatory changes, such that related lymph nodes may appear hypermetabolic that are not involved with tumor. In radiation treatment planning, it may not completely matter that hypermetabolic lymph nodes be proven to contain tumor if they represent a potential route of spread, either by tumor cells or by an inflammatory response, as long as they are reported and communicated to the radiation oncologist to be included in the radiation field. Obviously, enlarged or morphologically abnormal nodes, asymmetrical enlargement or metabolically active nodes without an obvious inflammatory cause, necrotic nodes or those with indistinct margins are more likely to be malignant (Figure 2). However, based on SUVmax alone, 50% of malignant nodes fall in the range of reactive nodes, either by the whole body FDG PET-CT protocol or the dedicated high-resolution head and neck protocol [3]. Unless nodes exceed SUVmax of approximately 6.5 for the whole-body protocol, or 9.6 for the dedicated head and neck protocol (assuming it is performed after the whole-body protocol), the magnitude of metabolic activity cannot reliably distinguish between reactive and malignant nodes [3]. Using the head and neck protocol used by our center, we have reported the sensitivity, specificity, PPV and NPV for detecting positive regional nodes as 57%, 88%, 85% and 76% for contrast-enhanced CT, 70%, 82%, 57%, and 88% for the whole body FDG PET-CT protocol, and 91%, 71%, 51% and 96% for the high-resolution FDG PET-CT head and neck protocol. This reflects a typical trade-off between sensitivity and specificity for FDG PET-CT.

In assessing response to treatment, the most unequivocal outcome is complete anatomic and metabolic resolution of the pre-treatment tumor (Figure 3). Conversely, tumors that remain significantly hypermetabolic following treatment do not create diagnostic uncertainty (Figure 4). However, in assessing response to treatment, interobserver variability, lack of standardization of interpretive criteria, limited evidence of medicine to support specific PET-CT criteria that signify a positive and negative study, and the lexicon for interpreting and reporting treatment response have hampered FDG PET-CT utility in head and neck cancers. This has also limited the ability to compare studies between institutions. To address this, interpretive criteria have been proposed by consensus groups, including the Hopkins semiquantitative criteria and the American College of Radiology (ACR) qualitative consensus criteria, NI-RADS [6,7]. These criteria are shown in Table 1 and Table 2. For both, it is recommended that PET-CT be delayed until 12 weeks post treatment to avoid potential false positives due to post-treatment inflammatory change (Figure 5). NI-RADS categories for PET-CT include information for both FDG uptake and findings on contrast-enhanced CT, further supporting the value of a high-quality thin section contrast-enhanced CT, either separately or concurrently with the PET scan. NI-RADS has been shown to result in moderate inter-observer variability between both experienced and novice readers and a high negative predictive value (91% for category 1 and 85% for category 2) [8,9]. One limitation of NI-RADS is that it does not address a category of mild-moderate FDG uptake in a residual mass or lymph node than has decreased in size but is still enlarged. This is a common finding, and our recommendations are that, if biopsy is not performed, close follow-up is observed, preferably with PET-CT at an additional 2–3 months to ensure further regression or resolution.

NCCN endorses the use of FDG PET-CT in staging, therapy assessment and surveillance of head and neck cancers. As stated by the NCCN guidelines, “US, CT, MRI, and PET-CT all have unique advantages and disadvantages when used as surveillance imaging. There is evidence that FDG PET-CT may be the most sensitive of these modalities” [10]. Despite this, NCCN acknowledges that there is no proven benefit of continued routine surveillance with PET-CT when a scan at 3 months post treatment is negative and there is no clinical suspicion of progression or recurrence. Recurrences, when they do occur, are usually within the first 24 months, and disease-free status for 5 years is the assumption of a cure. Therefore, close clinical follow-up is necessary. Very late recurrences (beyond 7 years) are considered second primaries, even at the same anatomic location of the first primary tumor. Distant metastatic disease is most typically to the lungs.

In head and neck cancers, the post-treatment false positive rate of FDG PET-CT is high, due to several causes including post-treatment inflammatory changes, fat necrosis, asymmetrical muscle activity post treatment (due to compensatory muscle activity or to inflammatory denervation), fistulas, and vocal cord paralysis (increased uptake in the vocal cord and cricoarytenoid muscle on the side opposite the paralysis) (Figure 6). Recurrent disease is often trans-spatial and does not follow normal anatomic planes (Figure 7). For evaluating response to definitive radiotherapy, the PPV of a positive PET-CT scan is low to moderate (33%, 88% nodal), the PPV of an equivocal PET-CT is low (6% primary, 33% nodal), the NPV of a negative PET-CT scan is high (95% primary, 91% nodal) [11]. Interim FDG PET-CT scanning (after one to three cycles of induction chemotherapy) may avoid inflammatory changes that occur post treatment and has been proposed [12]. However, there is a lack of standardized disease criteria for head and neck cancer response on interim FDG PET-CT, and different protocols for induction chemotherapy limit the value of this approach. Given the high false positive rate after treatment, the most rational use of FDG PET-CT is to identify suspicious lesions to direct biopsy or for directed monitoring. False negatives are rare for most head and neck cancers, but do occur. Some salivary gland tumors are relatively low in activity, as well as tumors with a low cellularity, such as clear cell histology [1]. Highly necrotic or cystic lymph nodes may be low in activity, although a rim of hypermetabolic tissue is typically, but not always, present.

Perineural spread of tumor is subtle on PET-CT. It can occur without motor or sensory symptoms and requires a refined knowledge of cranial nerve anatomy. It can be suggested by linear metabolic activity along the anatomic distribution of a nerve, or increased metabolic activity within a neuroforamen of the skull base, or within the trigeminal cistern (Meckel cave) (Figure 8). Intracranial extension of tumor may be difficult to identify on PET because of similar activity to the normal brain (Figure 9). Tumor in the pterygopalatine fossa signifies a high risk of perineural spread to the cavernous sinus and/or the semilunar ganglion within the trigeminal cistern [13]. Extension to the fossa is typically signified by obliteration of the fat and/or the presence of metabolically active tissue and is important to recognize (Figure 10). The suspicion of perineural spread is best confirmed by contrast-enhanced MRI [13].

In interpreting and reporting FDG PET-CT for head and neck cancer, several factors should be considered. Correct anatomic descriptions and a knowledge of the typical regional nodal spread from specific primary sites should be applied. Characterization of nodal features, particularly with respect to possible extracapsular extension has significant prognostic value (with the exception of HPV-associated oropharyngeal squamous cell carcinoma). Careful attention to vulnerable regions and evidence of perineural spread should be made, with recommendations for confirmatory MRI, when appropriate, and to determine whether intracranial extension has occurred. Finally, the inclusion of a head and neck radiologist adds value and credibility to the interpretation and the structuring of the report. 

### 2.2. Squamous Cell Carcinoma of the Head and Neck (SCCHN)

SCCHN is the most common head and neck malignancy and the sixth most common malignancy world-wide. It arises from the epithelium of the upper aerodigestive tract [14]. Tumors of the oral cavity are most typically associated with tobacco use, periodontal disease, and excessive alcohol consumption. SCCHN of the oropharynx is most commonly linked to oncogenic strains of HPV, particularly HPV-16 and HPV-18, which are now protected by HPV vaccinations [15]. HPV-positive tumors have a generally more favorable prognosis [16].

Treatment of SCCHN depends upon the location and stage [10,17]. Localized disease (Stage I-II) is often managed by surgical resection, with or without nodal dissection, or definitive radiation, if not a surgical candidate. Locally advanced disease (Stage III-IVB) is typically managed by concurrent chemotherapy and radiation therapy (CRT), with or without resection and nodal dissection. Induction chemotherapy may be given to reduce tumor bulk prior to radiation or surgery. SCCHN with distant metastatic disease (Stage IVC) is typically managed by single- or combined-agent chemotherapy. There are several basic principles that should be considered with respect to FDG PET-CT that relate to the primary site of known tumor.

Oral cavity squamous cell carcinoma

The oral cavity includes the lips, hard palate, floor of the mouth, the anterior two thirds of the tongue, the retromolar trigone, the alveolar ridge, and the buccal surface. Oral tongue SCC typically spreads to the muscles of the floor of the mouth and is typically markedly hypermetabolic on FDG PET-CT (Figure 11). First-order nodal drainage is typically to the submental and submandibular nodes (level 1), then to the jugulodigastric nodes (level 2). False positive FDG PET-CT scans can occur because of denervation changes post treatment (Figure 6). Bitten tongue, aphthous ulcers and/or other inflammatory conditions may produce false positive scans. Oral cavity SCC occurs on the alveolar surfaces, floor of mouth and the gingivobuccal and gingivolingual sulcus. With any oral cavity squamous cell carcinoma, it is important to evaluate for potential osseous invasion (Figure 12), and to assess for potential invasion into the maxillary sinus or from maxillary sinus tumors. FDG PET-CT is not routinely done for clinically N0 disease of the oral cavity. Periodontal abscesses and periodontal disease, aphthous ulcers, and osseous changes due to radiation necrosis, osteosarcoma, metastases, and invasion from maxillary sinus tumors should be carefully considered. Tumors of the oral cavity, sinuses and oral pharynx may spread via branches of the trigeminal nerve. Nodal spread initially to levels 1–3 is most common with oral cavity SCC [18].

2.Oropharyngeal squamous cell carcinoma

The oropharynx consists of the palatine (faucial) tonsils, the adjacent lateral pharyngeal wall, the soft palate, and the base of the tongue (lingual) tonsillar tissue. The tumor and involved nodes are typically intensely hypermetabolic on FDG PET-CT (Figure 13 and Figure 14). Spread of tumor is typically laterally to the parapharyngeal space and inferiorly to the vallecula and preepiglottic space [19]. Level 2 jugulodigastric nodes are most commonly involved, and there may be spread to level 3 or retropharyngeal nodes. As for the oral cavity, direct spread from the oropharynx can also occur to the pterygopalatine fossa with perineural involvement. Detection of the primary tumor when patients present with squamous cell carcinoma of a cervical lymph node in the absence of a known primary site of disease is a challenge. FDG PET-CT can identify the primary lesion in over half of these patients, and is associated with improved survivals [20]. If the nodal disease is HPV-related, the most likely primary site is the oropharynx [21].

3.Hypopharyngeal squamous cell carcinoma

Most hypopharyngeal squamous cell carcinomas are clinically silent until advanced. Five-year survival with small (T1-2) lesions is about 60% but all-stage 5-year survival is only 30% [22]. The hypopharynx is bounded superiorly by the oropharynx (at the level of the hyoid bone) and inferiorly by the cervical esophagus (at the lower margin of the cricoid cartilage). The sub regions of the hypopharynx include the pyriform sinuses (paired), the post cricoid region and the posterior pharyngeal wall (from the hyoid to the bottom of the cricoid). Tumors are typically markedly FDG avid (Figure 15). Spread of tumor from the posterior pharyngeal wall typically extends first to the retropharyngeal space, then to the prevertebral space, with increased incidence of pathologic retropharyngeal lymph nodes. SCC of the post-cricoid region is often insidious in onset. Tumor arising in the post-cricoid region may spread anteriorly to the cricoid cartilage and larynx, superiorly to the base of tongue and inferiorly to the cervical esophagus. Spread of tumor from the pyriform sinuses often extends laterally to involve the thyroid cartilage, lateral soft tissues, and anterior cervical neck nodes, the latter often being clinically occult [23]. Imaging is critical is management of hypopharyngeal tumors, with critical factors being extension to the base of the tongue, the cervical esophagus, posterior prevertebral space invasion, and cricoid cartilage invasion. Although identification of nodal disease is important, 60% of early T-stage hypopharyngeal tumors have microscopic disease in lymph nodes, unlikely to be detected by any imaging modality. FDG PET-CT can be helpful in regional staging but is particularly useful in detecting distant metastases, which can occur frequently. FDG PET-CT can also be helpful in excluding malignancy with asymmetrical pseudo-lesions of the pyriform sinus. The differential diagnosis of hypopharyngeal SCC includes aberrant cervical thymic tissue, lymphoma (with typically less necrotic nodes), Kaposi’s sarcoma and adenoid cystic carcinoma of the minor salivary gland tissue. 

4.Cervical esophageal squamous cell carcinoma

Cervical esophageal cancer is typically SCC and is defined as occurring from the lower margin of the cricoid to the thoracic inlet. It comprises only 2–10% of esophageal cancers, with dominant risk factors of tobacco and excessive alcohol intake. The 5-year survival is 30%. Because of local invasion into critical structures by the time of detection, surgery is often not feasible and combination CRT is often the therapy of choice for curative intent. Cervical esophageal SCC spreads by direct extent posteriorly to the retropharyngeal and then prevertebral spaces, anteriorly to the trachea and thyroid (visceral space), and laterally to the carotid space. Nodal metastases are typically to Level 4 and the superior mediastinum (Figure 16). As for other esophageal tumors, superficial or small tumors may be difficult to identify by FDG PET-CT. Endoscopic ultrasound is best for assessing depth of tumor (T stage).

5.Laryngeal squamous cell carcinoma

Laryngeal SCC, which is the most common laryngeal neoplasm, involves 3 subsites, the supraglottic larynx, the glottic larynx (glottis) and the subglottic larynx. Because of relatively poor vascularity and lymphatic drainage of the glottis and subglottic larynx, nodal and distant metastases occur late and local invasion is the primary pattern of initial spread.

 a.Supraglottic larynx

The supraglottic larynx extends from the hyoid superiorly to the superior aspect o the vocal cords inferiorly, and includes the false cords, the aryepiglottic folds, epiglottis, deep pre-epiglottic space and paraglottic spaces (Figure 17). Tumors of the supraglottic larynx tend to be larger than glottic or subglottic tumors and may involve hyoid bone and laryngeal cartilage, with sclerosis or destruction, as well as lymph node involvement more in keeping with patterns of spread of pharyngeal tumors. Direct invasion is to the pre-epiglottic space, paraglottic space and pyriform sinuses/hypopharynx. FDG PET-CT findings include hypermetabolic asymmetry although epiglottic tumors may be symmetrical, side-to-side (Figure 18). Tumors of the aryepiglottic folds may spread posteriorly to the hypopharynx. Breathing, swallowing and coughing can complicate FDG-PET assessment. PET-CT is particularly valuable in post-laryngectomy patients and in the pre-epiglottic space, where endoscopic evaluation is limited [24]. PET-CT may also be useful in differentiating tumor from a laryngocele of the supraglottic larynx, which, when fluid-filled, may simulate a necrotic mass.

 b.Glottic larynx

Glottic laryngeal SCC arises from the mucosal surface of the true vocal cords. Typical FDG PET-CT findings include asymmetrical hypermetabolism with a mass (which may be minimal in early stage). The tumor may extend across the anterior commissure, which is a typical pattern of spread (Figure 19). Pitfalls may include unilateral recurrent laryngeal nerve palsy, which can result in hypermetabolism of the contralateral cord and cricoarytenoid muscle (Figure 20). Talking should be discouraged during the interval from injection to imaging. Coughing and clearing of the throat should be suppressed to reduce physiological uptake of FDG. Sclerosis and hypermetabolism typically signifies cartilage involvement. Other laryngeal conditions that may mimic laryngeal SSC and demonstrate variable degrees of hypermetabolism on FDG PET-CT include lymphoma (typically enhancing and non-necrotic), rheumatoid arthritis of the larynx, sarcoidosis, chondrosarcoma of the thyroid cartilage or hyoid, laryngeal paraganglioma and adenoid cystic carcinoma of minor salivary tissue in the larynx (rare). Vocal cord paralysis treated with silicone implantation is typically hypermetabolic on FDG PET-CT, as can be seen with a foreign body granuloma reaction, especially with Teflon pledget placement.

 c.Subglottic larynx

The subglottic larynx extends from the inferior true vocal cord surfaces to the lower border of the cricoid cartilage. SCC of the subglottic larynx is relative rare. Spread can occur anteriorly through the cricothyroid membrane into the thyroid gland, posteriorly through the cricoid cartilage and into the hypopharynx, cephalad to the true cords and supraglottic larynx, and inferiorly to involve the cartilage rings and occlude the tracheal lumen. The differential diagnosis is similar to that for the glottic larynx, and FDG PET-CT may be positive to varying degrees in all of these entities (Figure 20). Airway obstruction is a common presenting feature. 

### 2.3. Nasopharyngeal Carcinoma

There are a variety of malignant primary tumors that occur in the nasopharynx. Lymphoma of the nasopharynx may occur rarely, primarily of NKTCL-nasal type, and diffuse large B-cell lymphoma [25]. Rhabdomyosarcoma, minor salivary tumors, and extension of tumors from the paranasal sinuses may also involve the nasopharynx. However, the most common malignant tumor of the nasopharynx in the adult is nasopharyngeal carcinoma (NPC) [26,27]. NPC is a distinct type of squamous cell carcinoma. It occurs as several subtypes. Keratinizing type (Type 1) occurs typically in the US. Non-keratinizing (Type 2) is commonly found in SE Asia. An undifferentiated subtype also exists. Black North Africans are at high risk, as well as endemic areas in SE Asia. NPC disease is three times more common in males than females. EBV, possibly HPV, dietary factors and smoking have been implicated in the etiology. 

NPC typically arises in the lateral pharyngeal recesses and extends within the nasopharynx and laterally into the parapharyngeal fat, and retropharyngeal nodes, Level 2 and 5 nodes, with other cervical nodal groups involved later [28]. Distant metastases are most common to the bones, followed by the lungs, liver and distant nodes [29]. NPC is typically moderate to high in metabolic activity (refer back to Figure 3). Diagnosis of NPC is most typically made by nasal endoscopy and CT. FDG PET-CT has been shown to be of value in the initial staging of NPC, where sensitivity and specificity have been reported to be 83% and 97%, respectively [30]. Lymphoma and minor salivary tumors in the lateral pharyngeal recesses can mimic nasopharyngeal carcinoma. Tumors arising from the sinuses and spreading to the nasopharynx may also mimic NPC. FDG PET-CT is widely used in patients with NPC with superior detection of nodal disease (especially retropharyngeal nodes) and distant metastases when compared to other modalities, as well as an increasing role in radiation treatment planning [31]. 

Treatment of NPC is dependent on stage. Radiation may be used alone in early-stage disease, or with added chemotherapy, either concurrent or sequential, in advanced cases. Treatment of progressive or recurrent disease typically includes check-point inhibitors [32]. In evaluating residual or recurrent disease post treatment, FDG PET-CT has been shown to exceed MRI in performance [33]. 

### 2.4. Sinonasal Tumors

Sinonasal tumors include rare neoplasms of the both the nasal cavity and paranasal sinuses. Because both sites are contiguous and typically involved, the tumors are often considered together. There are over 70 types of sinonasal tumors, both benign and malignant. The categories of sinonasal carcinoma differ in terms of histologic, imaging, and clinical features [34,35]. A complete discussion of all but the more common sinonasal malignancies is beyond the scope of this report. The more common types of sinonasal malignancies to be addressed here include sinonasal squamous cell carcinoma, sinonasal adenocarcinoma, sinonasal undifferentiated carcinoma and esthesioneuroblastoma. Lymphoma, particularly NKTCL-nasal type, rhabdomyosarcoma, mucosal melanoma and sinonasal salivary tumors and extramedullary myeloma are also occasionally seen. 

FDG PET-CT has proven useful in staging, therapy assessment and restaging of sinonasal tumors. Some have suggested that the diverse metabolic signatures on FDG PET may characterize different types of sinonasal tumors [35,36]. The optimal timing of FDG PET-CT to identify recurrence in sinonasal carcinomas has been suggested as between 1–3 months, and >18 months, based on a bi-modal likelihood of recurrence [37]. However, post-treatment inflammation may complicate early assessments of treatment response. Although initial post-treatment PET-CT imaging is typically performed at 10–12 weeks post treatment, delayed imaging at up to 24 months may show further resolution of uptake due to post-treatment inflammation [38]. 

Locally invasive sinonasal tumor may extend into the skull base, orbit, or by perineural spread to the pterygopalatine fossa and/or intracranially. FDG PET-CT does not replace MRI as the preferred method for assessing for these complications [39]. The role of FDG PET-CT in sinonasal carcinomas has not been as well studied as other malignancies, because these tumors are rare. However, FDG PET-CT has been shown to be of value in detecting regional lymph nodes and distant metastases [40]. Clinically relevant information may be provided by FDG PET-CT in almost 50% of patients with newly diagnosed sinonasal malignancy [41]. It may also be of value in the evaluation of difficult to examine regions, such as the skull base and as a surveillance tool post treatment [42].

Sinonasal squamous cell carcinoma

Although the most common type of sinonasal malignancy, sinonasal (SNSCC) is nonetheless a rare entity with diverse subtypes and variations [43]. HPV and inverted papillomas may be pathogenically associated with SNSCC. SNSCC is typically high in metabolic activity on FDG PET-CT (Figure 21). For this reason, some have suggested that the distinction between benign and malignant degeneration of sinonasal papillomas can be made by PET [44]. However, benign inverted papillomas have been described with intense uptake on FDG PET-CT, making the differentiation between papillomas and squamous cell carcinomas by PET unreliable [45]. The nasal antrum is commonly involved with SNSCC, which can be poorly defined. The maxillary sinus is another common site. Invasion and bony destruction are common. 

2.Sinonasal adenocarcinoma

Sinonasal adenocarcinoma (SNAC) is a rare tumor arising from epithelial surfaces and seromucous glands of the nasal cavity and sinuses. There are multiple morphological, immunohistochemical and molecular subtypes. There are two major categories of SNAC: salivary type and non-salivary type. Salivary-type SNAC tumors, the most common of which is adenoid cystic carcinoma (ACC), are discussed below with other salivary tumors. Non-salivary-type tumors include non-intestinal and intestinal types. Non-intestinal types are divided into high- and low-grade tumors, with the prognosis of high-grade tumors being relatively poor. Although having a generally more favorable outcome than other sinonasal malignancies, SNAC involvement of the sphenoid sinus, invasion of the skull base, advanced T-stage tumor and unfavorable histology worsen prognosis [46]. Treatment of early-stage disease is typically by surgical resection and radiotherapy, but not typically chemotherapy unless there is distant metastatic disease [47].

3.Sinonasal undifferentiated carcinoma

Sinonasal undifferentiated carcinoma (SNUC) is an aggressive sinonasal tumor arising from the sinonasal epithelium. The male:female ratio of SNUC is 3:1, with a widely disparate 5-year survivals varying from 6.25 to 74%, likely because of the low number of reported series [48]. Histologically, undifferentiated features with neuroendocrine differentiation are seen. SNUC result in higher SUV values on FDG PET-CT than other sinonasal tumors [49]. SNUC has a propensity for local invasion and to metastasize widely and early (Figure 22). For this reason, FDG PET-CT may be of particular value in staging SNUC. Treatment is typically by multimodal approaches, such as surgery and radiation, or more recently, with induction chemotherapy followed by concurrent chemoradiation for responsive cases. Although surgery was variably used initially for SNUC, more recent data supports that it be reserved for cases unresponsive to other local treatment [49]. 

4.Esthesioneuroblastoma

Previously, esthesioneuroblastoma (ENB, also called olfactory neuroblastoma) and SNUC were thought to be variations of the same tumor. However, they are now considered distinct entities. ENB arises from the olfactory neuroepithelium in the upper nasal cavity. There are two separate classification systems for ENB, which classify the tumors by stage (Kadish system) and pathologic features and grade (Hyams system) [50,51]. There is a wide variation in the magnitude of metabolic activity in ENB. Unlike other head and neck tumors, magnitude of metabolic activity on FDG PET-CT is not necessarily proportionate to grade. ENB typically arise near the cribriform plate, frequently in the ethmoid sinuses, and have a propensity to invade into the floor of the anterior cranial fossa and/or into the orbits. Because of variable metabolic activity that may be similar to that of the brain, it may be difficult to identify intracranial or orbital involvement of ENB by PET alone and MRI or contrast-enhanced CT are typically required (Figure 23). Survival from ENB depends on the extent of initial disease. Treatment strategies are evolving but are typically multimodality [52]. Although the magnitude of uptake of FDG in ENB is a limitation in assessment of nodal disease and distant metastases, PET-CT is most critical in evaluating spread of ENB to retropharyngeal nodes [53]. Because ENB expresses somatostatin receptors, ^68^Ga or ^64^Cu DOTATATE PET-CT may be an alternative to imaging EBN and may support the possibility of future clinical theranostic trials for EBN that target the somatostatin receptor [54,55]. 

### 2.5. Major and Minor Salivary Gland, Lacrimal Gland Tumors

There are a large number of benign and malignant salivary tumors that represent a challenge for imagers. Metabolic activity on FDG PET-CT is variable, although most are high in activity [56]. The basic problem is that both benign and malignant tumors of salivary glands are prominent metabolically. It is not reliably possible to differentiate between these categories based on metabolic activity alone. Some tumors are unique to the parotid gland, which has some unique embryologic features, and others may involve the parotid as well as minor salivary glands and salivary tissue lining the buccal surface, soft palate, tongue, and upper and lower airways. Tumors that affect salivary glands can also involve the lacrimal glands, which have features in common with salivary tissue. However, 70% of salivary tumors, both benign and malignant, occur in the parotid glands. Lymph nodes within the parotid gland represent first-order drainage for many cutaneous malignancies of the periauricular skin, and metastatic lymphadenopathy from many other types of tumors. The parotid also represents a site of primary lymphadenopathy and includes non-Hodgkin lymphoma (Figure 24), benign lymphoepithelial lesions of HIV (BLL-HIV), and infectious causes. Parotid bed tumors also have a propensity to develop perineural spread, for example along the facial nerve and/or the auriculotemporal nerve (refer back to Figure 8) [57]. The most common benign salivary tumors are pleomorphic adenoma (benign mixed tumor) and Warthin tumor. The most common malignant salivary tumors are adenoid cystic carcinoma and mucoepidermoid carcinoma. Most benign and malignant salivary tumors as well as reactive and metastatic lymph nodes are hypermetabolic of FDG PET-CT. However, FDG PET-CT has been shown to be of value, both in initial staging and ongoing monitoring of the malignancies of salivary origin [58].

Pleomorphic adenoma and carcinoma ex pleomorphic adenoma

The most common salivary tumor is the benign pleomorphic adenoma (benign mixed tumor, or BMT). These are most typically found in the parotid gland and account for 65% of parotid tumors [56]. Pleomorphic adenomas are typically solitary and have prominent metabolic activity on FDG PET-CT, although the larger ones can have lower metabolic activity (Figure 25). These are typically resected because carcinoma ex pleomorphic adenoma (CXPA) can arise in up to 13.8% of pleomorphic adenomas, with an incidence that increases over time [59]. The magnitude of metabolic activity cannot suggest whether malignant degeneration into CXPA has occurred. Other imaging characteristics must be considered in this regard and the diagnosis is ultimately made by pathology. CXPA is an aggressive malignancy with a generally poor prognosis. In addition to histologic grade, other factors that affect outcome of CXPA are T-stage, perineural involvement and nodal disease. Treatment is by radical, en bloc resection followed by radiation [60,61]. Most CXPA occur in the parotid gland, but may also occur in minor salivary glands. Pleomorphic adenoma accounts for 5–25% of lacrimal orbital tumors [62] and CXPA is also the second most common epithelial malignancy to involve the lacrimal gland as well [63] (Figure 26). 

2.Warthin tumor

Also known as papillary cystadenoma lymphomatosum, Warthin tumor is the second most common salivary tumor, typically occurring within the parotid gland in older male smokers. These tend to be unifocal and are hypermetabolic on FDG PET-CT (Figure 27). Although MRI and ultrasound features may be suggestive of Warthin tumor, diagnosis typically requires biopsy [64,65].

3.Mucoepidermoid carcinoma

Mucoepidermoid carcinoma (MEC) is the most common salivary gland malignancy, typically occurring in the parotid but also within minor salivary glands and tissues, including the oral cavity, oral pharynx as well as a similar tumor within in the bronchi [66,67,68]. Although typically a lower-grade lesion, higher-grade forms do occur. The histology is one of a mixed lesion, with mucous, epidermoid and intermediate cell types. Metastatic spread is typically initially to lymph nodes. The majority of the literature related to the value of FDG PET-CT in MEC is focused on bronchial MEC, where it has been shown to be of value in predicting tumor grade and long-term prognosis [69]. The literature related to FDG PET-CT in other sites of MEC is sparse. As with adenoid cystic carcinoma, MEC can also have late recurrences, and with our anecdotal experience, this may occur with metastatic disease and a high-grade histology (Figure 28) [70].

4.Adenoid cystic carcinoma

Adenoid cystic carcinoma (ACC) is a rare malignancy of the sinonasal region and salivary glands. It is often insidious in onset but with a high metastatic potential. It is infiltrative and has a propensity for perineural spread and skull base involvement. Metabolic activity on FDG PET-CT is typically only mild-to-moderate, so that identification of intracranial extension requires CECT or MRI (refer back to Figure 9). In addition to the sinonasal region, ACC most commonly occurs in the parotid gland, minor salivary glands, and in the submandibular gland but can occur anywhere there is salivary tissue, including the soft palate, buccal surfaces and trachea (Figure 29) [71,72]. ACC has also been rarely reported to de-differentiate into CXPA [73]. Negative surgical margins are difficult to achieve and adjuvant radiation is often employed post-surgically [74]. Even with complete initial resection, ACC can recur late, often decades after initial diagnosis. Therefore, long-term monitoring is necessary. FDG PET-CT has been shown to be of value in staging and monitoring for ACC and is superior to MRI in this regard, with a reported sensitivity and specificity of 94% and 89%, respectively [75]. PSMA-targeting PET ligands used for prostate cancer imaging have high uptake in salivary glands. They have also been found to have increased uptake in ACC metastases and may have some future use as a theranostic strategy [76].

### 2.6. Thyroid Cancer

Thyroid Incidentaloma

It is relatively common to observe incidental uptake of FDG in a thyroid nodule on a PET-CT scan done for unrelated reasons (incidentaloma). The incidence of thyroid cancer in hypermetabolic nodules has been reported to be between 21.4% and 43.8% [77,78]. The American Thyroid Association supports that FNA should be performed for metabolically nodules > 1 cm in diameter (Figure 30) [79]. Hypermetabolic nodules < 1 cm can be monitored by ultrasound. Diffuse uptake of FDG in the thyroid is typically due to chronic lymphocytic thyroiditis (Hashimoto’s thyroiditis) and should prompt thyroid function tests since many of these patients will be hypothyroid. If there is clinical evidence of Hashimoto’s, it does not require further evaluation. Multinodular goiter is typically associated with nodules of varying degrees of metabolic activity. Grave’s disease is typically not associated with increased metabolic activity in the thyroid. However, it should be recognized that intensely hypermetabolic, markedly enlarged thyroid may represent thyroid lymphoma. Diffusely hypermetabolic, infiltrative thyroid masses could also represent anaplastic thyroid cancer. 

2.Differentiated thyroid cancer

The role of FDG PET-CT in the management of differentiated thyroid cancer (DTC), either papillary, follicular, or mixed, is well-accepted in the restaging of suspected persistent or recurrent non-iodine avid DTC [80,81]. In this regard, several clinical scenarios justify FDG PET-CT. These include a non-iodine avid target lesion, such as a lung nodule or bone lesion, or an elevation in thyroglobulin or antithyroglobulin antibody in the absence of a known site of recurrence by conventional imaging or whole body I-131 scan [82]. Under these circumstances, FDG PET-CT will typically demonstrate recurrent or residual sites of DTC tumor involvement as being hypermetabolic (Figure 31). FDG PET-CT is also important in evaluating for retropharyngeal nodal involvement, which is often clinically silent and difficult to recognize by other imaging modalities. There is some evidence that the sensitivity of FDG PET-CT for DTC is greater if the thyroid stimulating hormone (TSH) is elevated at the time of imaging, although the evidence to support the value of exogenous human recombinant TSH (rhTSH) in PET-CT for DTC is contradictory [83]. In patients with high-risk differentiated thyroid cancer, there is some support for the application of FDG PET-CT as an initial staging procedure [84]. It should be noted that differentiated and undifferentiated thyroid cancer can be positive on PSMA-targeting PET-CT that is done for prostate cancer [85]. This may support a future theranostic target in patients with treatment-refractory DTC or anaplastic thyroid cancer [86]. Incidental focal uptake of either FDG or a PSMA-ligand in the thyroid or in cervical nodes in a patient with prior thyroid cancer should prompt additional evaluation.

3.Anaplastic thyroid cancer

Anaplastic thyroid cancer (ATC) is one of the most aggressive of all tumors. Prognosis is poor, with an historical disease specific mortality of 98–99% and a 6-month survival of 50% [87]. Treatment of early-stage disease is typically by surgery and combination CRT, which are intended for palliation, and may improve quality of life and slightly prolong survival. However, clinical trials that leverage molecular features to enable targeted therapies are underway [88,89,90]. ATC primary tumors and their metastases are typically intensely hypermetabolic on FDG PET, which is effective in defining extent of disease (Figure 32) [91]. Despite the poor prognosis of ATC, FDG PET-CT has been reported to affect management in up to 50% of patients with ATC [87]. FDG PET-CT may also provide a theoretical mechanism to monitor response to novel targeted therapies.

4.Medullary thyroid cancer

Medullary thyroid cancer (MTC) accounts for 5% of thyroid malignancies, and arises from the C cells of the thyroid, which produce calcitonin. Serum calcitonin is a marker of disease activity and recurrence. Seventy-five percent of cases of MTC are considered sporadic, with the remainder associated with multiple endocrine neoplasia type 2 (MEN2), an autosomal dominant condition mediated by the RET proto-oncogene [92]. Staging of MTC is typically done by conventional imaging. The disease is typically managed by attempted complete surgical resection. Prophylactic thyroidectomy may be done in at-risk patients. Post-surgical radiation is used in some circumstances. Tyrosine-kinase inhibitors have been leveraged successfully for disease control in patients with metastatic disease, with cytotoxic chemotherapy reserved for patients with refractory disease [93]. MTC is typically hypermetabolic on FDG PET-CT (Figure 33). The best supported use of FDG PET-CT is in treated MTC patients with persistent or recurrent elevations in calcitonin. Rapid doubling times (<12 months) and higher calcitonin levels (>1000 pg/mL) improve performance of FDG PET-CT, with a PPV of 88% [94]. However, the relatively low NPV (57.7%) and a per-patient detection rate of only 59% of FDG PET-CT in findings sites of suspected recurrent MTC limits its utility. Successful PET imaging of MTC using somatostatin-binding or PSMA-binding analogs has been reported [95,96,97,98]. However, superiority of those methods over FDG PET-CT has not been proven, although DOTATATE PET may be somewhat more sensitive for bone metastases than FDG PET-CT [97,98].

## 3. Neuro Oncology

### 3.1. Technical Notes and Overview

Over the past several decades there has been a great deal of information published using PET to assess brain and central nervous system (CNS) tumors. Numerous radiopharmaceuticals have been developed to explore the biology of these tumors. Despite these ongoing efforts only one radiopharmaceutical, ^18^F-fluordeoxyglucose (FDG) has been FDA approved for brain PET. Nowadays, the role of FDG PET in evaluation of brain tumors is best considered as a problem-solving modality, secondary to conventional imaging modalities, such as MRI or contrast-enhanced CT. Nonetheless, there has been an expanding role of PET-CT in neuro oncology. In addition to FDG, several radiopharmaceuticals have been developed and used to assess important biologic and physiologic properties of brain tumors including metabolism, proliferation, blood flow/perfusion, hypoxia, amino acid uptake, somatostatin-receptor expression [99,100,101,102,103,104,105]. Expanded neuro oncologic applications for a variety of radiopharmaceuticals, including those FDA-approved for other indications, is well supported in the literature. However, their widespread clinical use has been limited by lack of reimbursement for the brain tumors. It is anticipated this barrier will gradually be lifted over time and additional supportive evidence. 

FDG is an analog of glucose, being actively transported across the blood–brain barrier (BBB) and is the primary source of energy in normal brain and brain tumors. The brain shows FDG uptake to different degrees in the grey matter, white matter, and ventricular system. The primary confounding factors in using FDG to assess brain tumor metabolism is the relatively high normal physiologic FDG gray matter uptake which reduces the contrast between tumor and normal brain. The resolution of PET imaging is also problematic for small lesions. Co-registration of FDG and MRI images acquired separately, or with dedicated PET-MR scanners may allow for a greater ability to identify small or treated lesions. This also allows for improved determination of residual viable vs. treated sites of tumor. Improving image quality may also be achieved by delaying image acquisition to accentuate differences between the tumor and normal brain tissue [103,106,107]. 

### 3.2. Paraneoplastic CNS Manifestations and Sources of False Positive FDG PET Scans That Can Mimic Brain Tumors

A confounding factor of the use of FDG PET-CT in identifying brain tumors, either primary or metastatic, is the fact that focal cerebral hypermetabolism is seen in a variety of conditions. It is important that imaging professionals and referring providers be aware of these conditions, which can mimic brain tumors or signal the presence of an underlying systemic malignancy with paraneoplastic neural manifestations [108]. Encephalitis is a devastating condition that can be seen with either paraneoplastic and autoimmune conditions as well as patients treated with check-point inhibitors [109]. The pathogenesis of these conditions may involve antibodies directed against neuronal components, such as those against the N-methyl-D-aspartate receptor (anti-NMDAR). These antibodies can be elevated in the serum and cerebral spinal fluid. Anti-NMDAR encephalitis may present with seizures, psychiatric disturbances, and cognitive decline. A variant of autoimmune or paraneoplastic encephalitis is limbic encephalitis, which typically involves the entire hippocampus with variable decrease metabolic activity more extensively in the temporal lobe (Figure 34). Anti-NMDAR encephalitis presents with patchy areas of cortical hypermetabolism. Metabolic activity in the posterior cerebrum is often diffusely decreased (Figure 35). No mass effect is present, although MRI FLAIR signal abnormalities may occasionally be present. 

Viral encephalitis can also present with increased metabolic activity without a space occupying lesion and may mimic primary brain tumors (Figure 36). MRI is critical for the diagnosis. Although not well studied, acute stroke can show increased metabolic activity within the penumbra of ischemic tissue, and subacute stroke may show increased uptake of FDG in areas of cortical laminar necrosis (pseudolaminar necrosis) [110] (Figure 37). Thus, a stroke in a patient with an underlying malignancy may mimic brain tumor on FDG PET-CT in the subacute phase. Although FDG PET-CT is often used in identification of an epileptogenic focus as an area of decreased metabolic activity, if injection occurs during an active seizure, the FDG PET scan will typically show regional hypermetabolism in regions affected by seizure propagation (Figure 38). Under these circumstances, active seizure activity can mimic underlying brain tumors. Again, MRI is critical in excluding this diagnosis. Immunotherapy and check point inhibitor therapy can result in encephalitis as well as hypophysitis, the latter showing increased uptake on FDG PET-CT in the pituitary gland [108,109,111]. Incidental note of uptake in the pituitary is most frequently due to either a micro adenoma (<1 cm diameter), or a macro adenoma (>1 cm in diameter) (Figure 39). However, the finding in a patient on immune check-point inhibitors should prompt further evaluation for hypophysitis. It should be reemphasized that, at the current time, the use of FDG PET-CT must be as a problem-solving modality secondary to conventional imaging approaches, such as MRI, and not as the single preferred modality. Nonetheless, it is important to recognize the many sources of false positive FDG PET scans that may mimic an underlying brain tumor or may present as paraneoplastic syndromes of underlying malignancy. 

### 3.3. Primary Brain Tumors

Di Chiro and coworkers published the first studies using FDG-PET to characterize gliomas [112,113]. High-grade gliomas were reported as exhibiting high FDG uptake and lower-grade gliomas less uptake. These initial studies were able to characterize survival times based on tumor uptake when compared to contralateral cortical FDG uptake. Since then, FDG PET has been applied with respect to primary brain tumors in several scenarios. Although MRI remains the mainstay of imaging brain tumors, PET-CT, as well as PET-MR, can provide information that can assist in solving several neuro oncologic imaging problems.

There are situations in which FDG PET-CT is performed specifically for questions regarding brain tumors. Two examples of this scenarios are to distinguish between CNS lymphoma and toxoplasmosis in the immunocompromised patient with ring-enhancing lesions, or as an imaging adjunct in distinguishing between recurrent glioma and the therapeutic effects resulting in pseudoprogression and radiation necrosis [114]. Although FDG PET-CT is being used effectively for treatment planning in head and neck tumors, it has not been shown to be of benefit in helping to define treatment volume in gliomas [115,116]. There are several PET radiopharmaceuticals that are not, at this point, FDA approved, but which show superiority in characterizing lesions in neuro oncology, including ^11^C-MET, ^18^F-DOPA, ^18^F-FET [105,117]. Although future clinical PET imaging may be expanded globally to incorporate these or similar radiopharmaceuticals for neuro oncology, the current FDA approved agent for PET brain imaging in the US is FDG. ^68^Ga and ^64^Cu DOTATATE shows promise in imaging meningiomas as well. ^18^F fluciclovine (Axumin^®^) and either ^68^Ga or ^18^F PSMA ligands used for imaging prostate cancer also demonstrate marked uptake in meningiomas. 18F fluciclovine has also been proposed to differentiate between low- and high-grade gliomas, with higher-grade lesions demonstrating greater uptake [118].

Primary brain tumors rarely spread to regions outside the brain. For this reason, primary brain tumors are typically graded rather than staged. Imaging, typically with MRI, is an important component of all brain tumor evaluation and management, including grading. However, FDG-PET has also been used in the grading of gliomas. FDG uptake is usually higher in high-grade glioma (WHO grade III and IV) than low-grade tumors (WHO grade II) [112,113,119,120]. Semiquantitative assessments of brain activity, such a comparison of activity in the tumor to the contralateral white matter may facilitate distinction between different grades of glioma and may have some prognostic significance [120,121]. However, it should be strongly emphasized that this is not predictably the case, and higher-grade gliomas can present with relatively low metabolic activity for unknown reasons (Figure 40). Therefore, the use of FDG PET-CT should be used with caution in attempting to grade gliomas. There is evidence that initiation of high dose corticosteroids, which are often used early after identification of brain tumors, suppresses global brain metabolic activity. However, whether high-dose corticosteroids may also suppress metabolic activity in high-grade gliomas has not been addressed. Nonetheless, high FDG uptake in known low-grade glioma often signifies transformation to a higher-grade tumor [122]. This can be leveraged to direct biopsy to the most concerning regions. In lesions without histologic confirmation, very high levels of metabolic activity (SUVmax > 10–15) on FDG PET are more likely to be primary CNS lymphoma (PCNSL) than gliomas [123,124]. For example, Yamashita et al. reported that the mean (±SD) of glioblastomas in their series of 37 patients was 13.1 ± 6.34, while that in a series of 19 patients with PCNSL was 22.5 ± 7.83 (*p* < 0.005). However, there was overlap in SUVmax between PCNSL and glioblastoma (previously known as glioblastoma multiforme) with maximum SUVs in the range of 15–20. 

Due to the heterogeneity and infiltrating nature of brain tumors, it can be difficult by either MRI or FDG-PET to define the exact geographic boundaries of the tumor. Defining the entire tumor volume is critically important in pre-surgical resection and pre-radiotherapy planning, assessing response to therapy, and in clinical follow-up and management [125]. MRI is the most commonly used modality in this regard. Contrast-enhanced MRI correlates with blood–brain-barrier (BBB) disruption and vascularity. FLAIR signal can be seen with edema and non-enhancing tumor. Low-grade gliomas typically do not enhance on MRI while high-grade gliomas enhance with gadolinium-based contrast agents. Unfortunately, there are reports of high-grade gliomas that do not enhance or display heterogeneous enhancement patterns on MRI [102]. By FDG PET-CT, there may be heterogeneous uptake including increased uptake in the white matter near the primary tumor. This can correlate with the enhancment and FLAIR signal when co-registration with FDG and MRI imaging.

### 3.4. CNS Lymphoma

Central nervous system involvement with lymphoma is relatively rare. There can be associated CNS involvement with systemic disease or primary central nervous system lymphoma (PCNSL), CNS involvement will often exclude patients for therapeutic trials. PCNSL is best evaluated with MRI. Nonetheless, FDG PET uptake in lymphoma is typically very high [126,127]. In a recent study using contrast-enhanced chest, abdomen, and pelvis CT and/or whole-body FDG imaging for initial staging, diagnostic yield was 2% and a false positive rate of 4%, for the detection of systemic lymphoma involvement. This study concluded that the diagnostic yield from whole body FDG PET imaging was low and not recommended [128]. A scenario in which FDG PET may be diagnostically helpful is in immunocompromised patients, such as those with AIDS, in whom enhancing lesions in the brain are found on MRI, particularly ring-enhancing lesions. In these patients, the primary differential considerations are CNS lymphoma vs. infection, particularly toxoplasmosis. Tumor and other infections may be associated with varying degrees of metabolic activity, but toxoplasmosis is typically hypometabolic on FDG PET [129] (Figure 41). However, there is a report of increased metabolic activity in a transplant patient with CNS toxoplasmosis [130]. Therefore, FDG PET-CT should be regarded as only one factor in the identification of the etiology of ring enhancing lesions, particularly in immunocompromised patients. As mentioned above, PCNSL is typically very hypermetabolic on FDG PET-CT (Figure 42). 

### 3.5. Brain Metastases

Brain metastases occur in approximately 20% of individuals who develop cancer. There is a wide range of metabolic activity on FDG PET-CT in brain metastases. FDG PET-CT is a standard body imaging modality in oncologic imaging. With increasing systemic and local treatment options such as immunotherapy, surgery, external beam radiation therapy, and stereotaxic radiotherapy, there has been improvement in the treatment in brain metastases [131]. MRI imaging is the modality of choice for initial and follow-up management of brain metastases [132,133]. FDG PET has a lower spatial resolution than MRI and has a sensitivity of about 20–30 percent [133,134]. Many metastatic lesions are not visible on FDG PET-CT because they display metabolic activity similar, or less than that of the normal brain (Figure 43). It should be emphasized that amino-acid PET-CT imaging has shown better performance than FDG PET-CT in clinical trials for brain metastases and is widely regarded as being preferable to FDG PET in imaging CNS metastases [134,135]. F-DOPA PET also shows promise in longitudinal assessment of brain metastases [136]. However, these approaches are not available in the US due to lack of FDA approval. Nonetheless, FDG -avid metastatic brain lesions may be appreciable during typical FDG PET-CT for systemic cancer evaluation and should prompt further investigation by MRI to define the full extent of disease (Figure 44 and Figure 45). Leptomeningeal metastatic disease is often inapparent by FDG PET-CT but peripheral foci of activity in the brain, or increased metabolic activity in the spinal canal, should be regarded as suspicious for leptomeningeal carcinomatosis and prompt MRI imaging for characterization and confirmation (Figure 46). Patients with evidence of peripheral perineural spread of tumor are also at risk for central CNS leptomeningeal involvement. Therefore, in patients with peripheral perineural involvement on FDG PET-CT, consideration should be given for MRI imaging of the neural axis.

### 3.6. Response to Therapy

FDG PET-CT has been shown to be of significant value as an adjunct to conventional imaging in evaluating response of brain metastases to treatment [139,140]. In this regard, it has been suggested that dual time-point imaging may provide better delineation of metastatic lesions, compared to background brain activity [140]. In treatment response and surveillance of CNS lymphoma, FDG PET-CT has been less well studied but may be of value, primarily in detecting recurrence [126,141]. In contrast, the use of FDG PET-CT in evaluating response to treatment of high-grade primary CNS gliomas has been extensively studied. 

Glioblastoma Multiforme (GBM) is the most common and fatal primary brain tumor. Earlier response assessment is vital in the management of GBM and all brain tumors. At present, current clinical practice relies upon conventional imaging to appraise the response to therapy. It remains a challenge to determine the treatment efficacy due to the lack of accuracy of conventional modalities in differentiating between tumor growth or recurrence and treatment-induced changes, such as pseudoprogression or radiation necrosis [103,142,143]. One of the most significant challenges of imaging the treatment response of primary brain tumors is distinguishing between radiation necrosis and recurrent tumor. MRI can be limited in this distinction and FDG-PET has been applied successfully in this scenario. By this technique, metabolic activity that is higher in the lesion of interest than in the contralateral normal white matter is considered at risk for recurrence (Figure 47), while that without metabolic activity is typical of radiation necrosis (Figure 48). This technique has also been used as a prognostic indicator [114,120,121,144,145]. However, the reliability of FDG PET as a prognostic predictor has been questioned by other studies [142,143,146,147,148,149,150]. Imaging methods to identify areas suspicious for recurrence may also be important in that biopsy confirmation is challenging in these patients because of significant intratumoral cellular heterogeneity [151,152]. Identifying the most suspicious area for biopsy by FDG PET aids in the diagnostic yield of biopsy.

### 3.7. Meningioma

Meningiomas are the most common intracranial brain tumors. Approximately 30% of intracranial tumors are meningiomas. The majority of meningiomas are benign (WHO grade I). They are slow growing and often stable over years, but may be treated by surgery if symptomatic. They have a low recurrence rate following complete resection. WHO grade II (atypical) and WHO grade III (malignant) meningiomas behave more aggressively with a significant recurrence rate of 40% in grade II and 80% in grade III tumors. FDG PET has been used for many years in the evaluation of meningiomas. An excellent review of PET imaging in meningiomas and other brain tumors has been recently published [104]. Studies have also been published showing the utility of FDG PET in differentiating low from high-grade meningiomas, with higher grade tumors showing increased uptake [108,153]. However, most meningiomas are either lower or similar to normal brain on FDG PET-CT and are difficult to appreciate on FDG PET-only images, an indicator that these are likely of a lower WHO grade (Figure 49). For this reason, SSTR (e.g. DOTATATE) PET-CT is preferred in for imaging meningiomas, with detection sensitivities greater than for MRI [104]. Most meningiomas are markedly positive on SSTR-targeting PET (Figure 50 and Figure 51), although it must be noted that Grade III, poorly differentiated malignant meningiomas may be decreased in uptake of SSTR ligands [154]. Therefore, a combination of modalities and PET radiotracers may ultimately be required to fully characterize residual, progressive or phenotypically malignant meningiomas. Meningiomas are frequently observed as having prominent uptake on PSMA and fluciclovine PET-CT although evaluation for brain tumors is not among the approved indications for these agents (Figure 52). Nonetheless, these new PET tracers may represent opportunities to identify recurrent meningiomas in cases where conventional imaging is equivocal or the region cannot be biopsied. 

## 4. Conclusions

FDG PET-CT is an essential modality in the evaluation of head and neck cancer (HNCa). The most successful imaging approach is one that maximizes image quality and ensures that the head and neck are immobilized for the study. In addition to a standard whole-body protocol, the separate acquisition of a high-resolution, thin-section, zoomed PET-CT with intravenous contrast best meets this challenge. The anatomy of the head and neck is complex. Expertise in being able to identify important small structures, and in understanding the regional significance, patterns of spread and distinguishing features of each of the types of HNCa is critical in maximizing the value of FDG PET-CT. In most cases, this is best achieved by the direct involvement of a neuro radiologist with expertise in head and neck imaging. 

Neuro oncologic applications of PET-CT, whether with FDG or some of the newer FDA approved agents, is an ever-evolving field. In most cases, PET-CT is best regarded as an adjunct to conventional imaging by MRI or CT. Maximization of the value of PET-CT in neuro oncology is best met in an interdisciplinary approach between neuro radiologists and nuclear medicine physicians/nuclear radiologists. A clear understanding of the patient history is critical to the correct interpretation of the images. Realistic expectations on the part of the referring providers caring for the patients requires a high level of communication with the imaging professionals. In summary, a team approach and good communication is critical in optimizing the value of PET-CT in HNCa and neuro oncology.

## Figures and Tables

**Figure 1 cancers-14-02726-f001:**
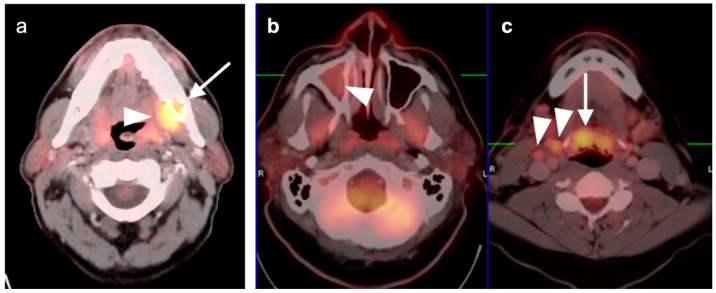
Inflammatory conditions of the head and neck that mimic tumor on FDG PET-CT. (**a**) Periodontal abscess, shown by a hypermetabolic inflammatory mass (white arrowhead) on the lingual side of an erosive osseous lesion of a tooth socket (white arrow); (**b**,**c**) right maxillary sinusitis, likely chronic because of neo osteogenesis (**b**, white arrowhead) with associated hypermetabolic lymphoid tissue in the base of the tongue (**c**, white arrow) and prominent hypermetabolic right cervical lymph nodes (**c**, white arrowheads).

**Figure 2 cancers-14-02726-f002:**
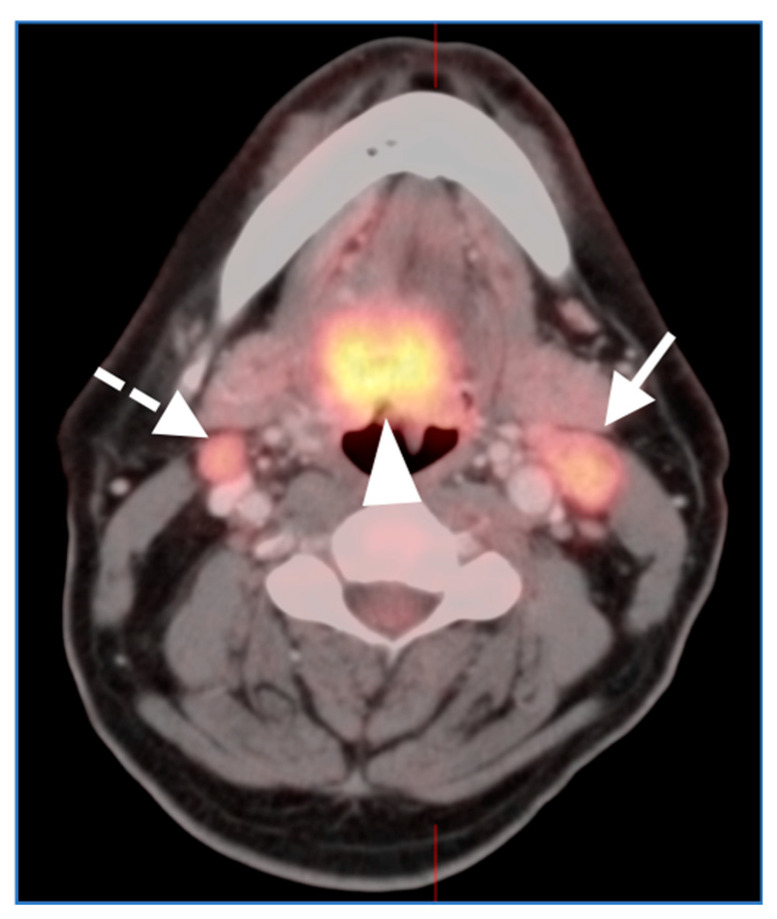
Squamous cell carcinoma of the central base of tongue with extension into the oral tongue (solid white arrowhead) with bilateral hypermetabolic lymph nodes. The left level 2A node (solid white arrow) is clearly large and heterogeneously enhancing, consistent with tumor involvement. The right sided node (dashed arrow) is smaller but similarly hypermetabolic, and was benign at biopsy. There is considerable overlap in metabolic activity between benign/reactive and malignant nodes.

**Figure 3 cancers-14-02726-f003:**
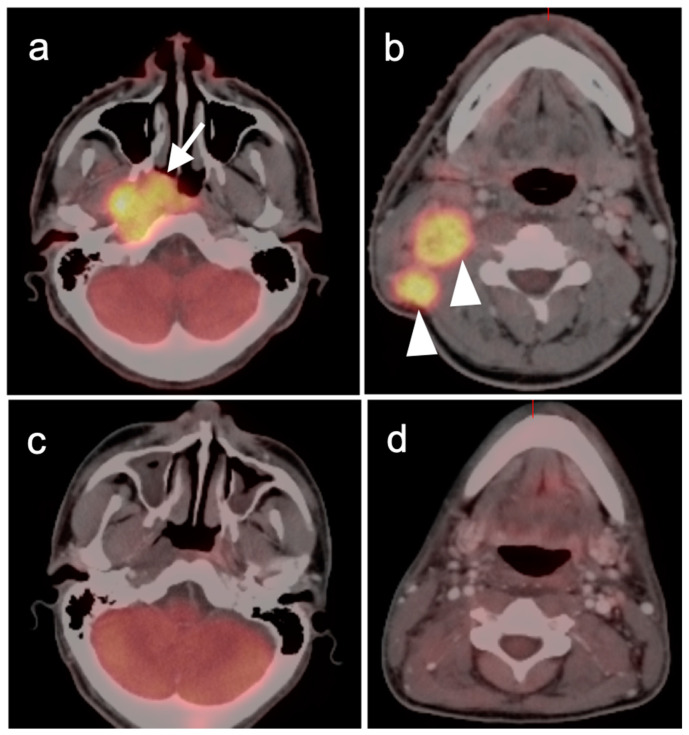
Complete metabolic and anatomic resolution of nasopharyngeal carcinoma. (**a**,**b**) Pre-treatment FDG PET-CT demonstrates hypermetabolic right nasopharyngeal mass (**a**, white arrow) and right pathologic cervical lymph nodes (**b**, white arrowheads); (**c**,**d**) post-treatment FDG PET-CT demonstrates complete anatomic and metabolic resolution of all sites of tumor.

**Figure 4 cancers-14-02726-f004:**
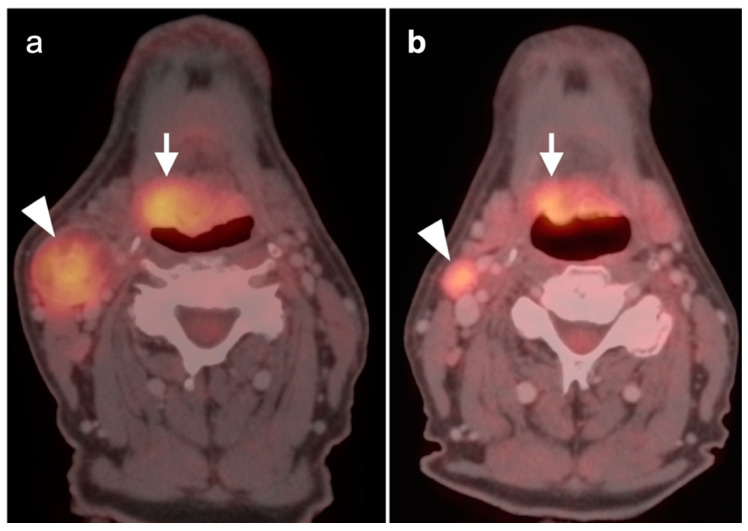
Incomplete response to treatment. Axial fused FDG PET-CT images before (**a**) and after (**b**) chemoradiation therapy demonstrate a right base of tongue squamous cell carcinoma (arrow) and ipsilateral right cervical pathological lymph node (arrowhead) with decreased, but still pathologic in both size and activity, after therapy.

**Figure 5 cancers-14-02726-f005:**
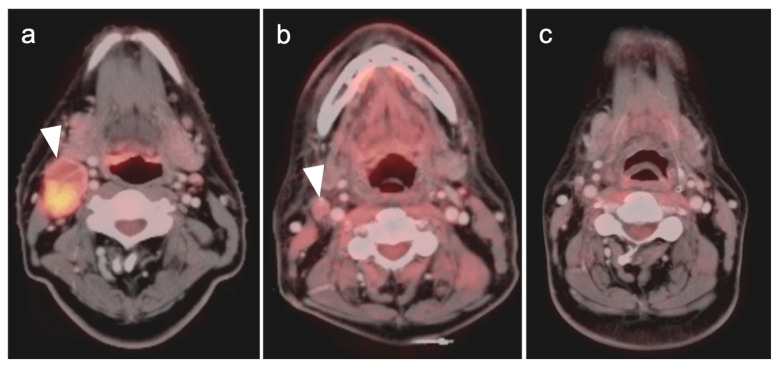
Early post-treatment inflammatory change. (**a**) Axial fused PET CT images prior to therapy demonstrate a hypermetabolic, enlarged pathologic cervical node (white arrowhead): (**b**) On the first post-treatment FDG PET-CT scan, the node has decreased in size and activity following but is still evident (white arrowhead) and has surrounding inflammatory changes and fat stranding; (**c**) Follow-up FDG PET-CT scan 3 months later shows complete resolution of the node and inflammatory stranding.

**Figure 6 cancers-14-02726-f006:**
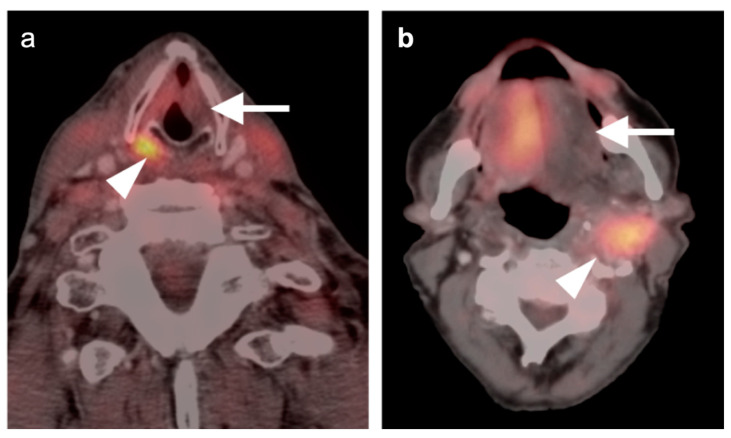
Muscle paralysis with contralateral increased metabolic activity in two patients. (**a**) Paralysis of the left vocal cord, which is patulous and low in metabolic activity (white arrow). There is compensatory contralateral increased activity in the right vocal cord and cricoarytenoid muscle (white arrowhead); (**b**) shown is a paralysis of the left hypoglossal nerve (CN 12) with low attenuation and hypometabolism of the left tongue musculature (white arrow), as well as compensatory increased activity in the right side of the tongue. Note area of recurrent tumor in the left neck (white arrowhead) along the expected course of CN 12.

**Figure 7 cancers-14-02726-f007:**
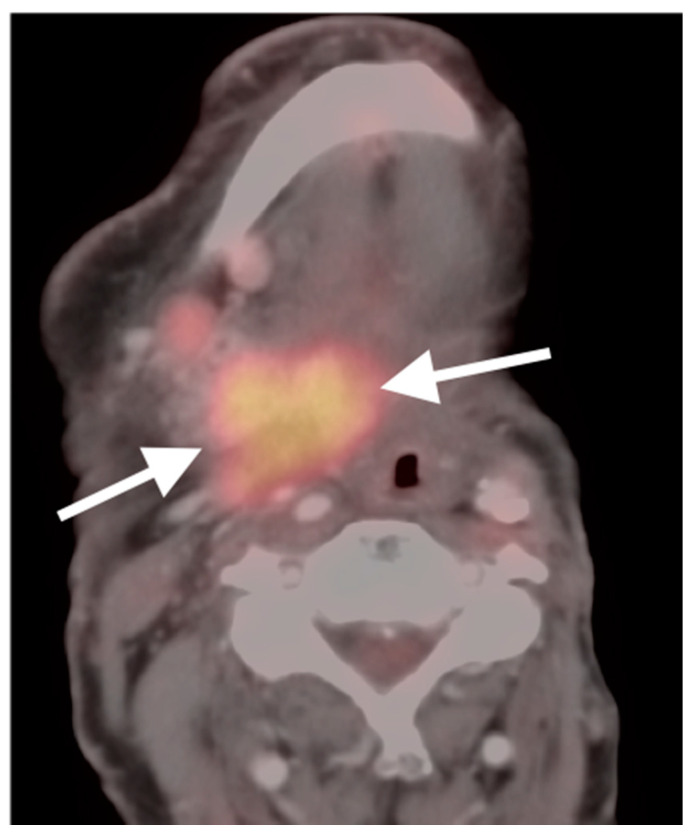
Trans-spatial recurrent tumor on FDG PET-CT (white arrows) in a patient who has undergone prior laryngectomy and left radical neck dissection.

**Figure 8 cancers-14-02726-f008:**
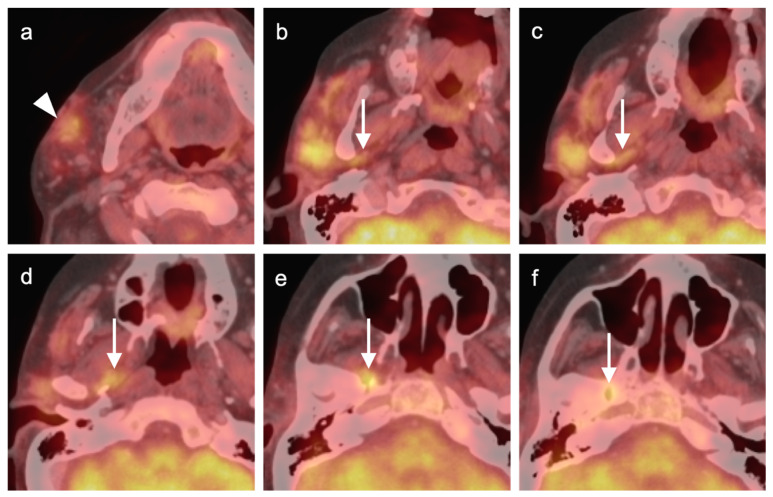
Perineural spread of tumor. (**a**) Hypermetabolic infiltrative tumor in the right parotid (white arrowhead, poorly differentiated carcinoma); (**b**–**e**) hypermetabolic tumor (white arrow) along the distribution of the right auriculotemporal nerve (branch of the mandibular nerve, V3); (**f**) extension of metabolically active tumor into the right foramen ovale (white arrow).

**Figure 9 cancers-14-02726-f009:**
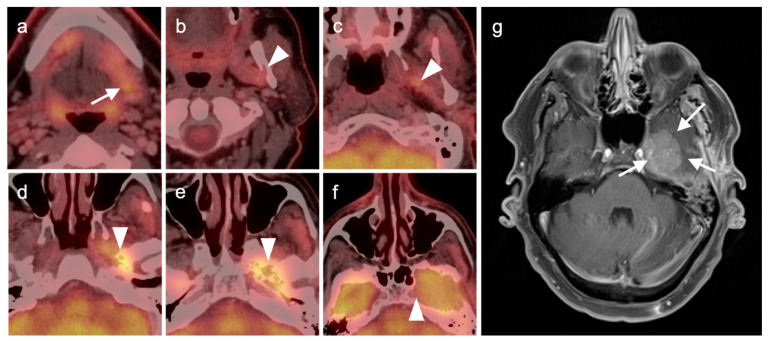
FDG PET-CT findings of perineural spread of tumor. (**a**) Small mildly hypermetabolic adenoid cystic carcinoma of the left sublingual gland (white arrow); (**b**–**e**) spread along the mylohyoid nerve, inferior alveolar nerve, and mandibular nerve (white arrowheads); (**f**) spread of tumor into the trigeminal cistern (Meckel’s cave) and into the middle cranial fossa, which is difficult to appreciate on FDG PET-CT (white arrowheads); (**g**) tumor is better defined by contrast-enhanced axial T1 with fat saturation MRI extending from the trigeminal cistern into the middle cranial fossa (**g**, white arrows).

**Figure 10 cancers-14-02726-f010:**
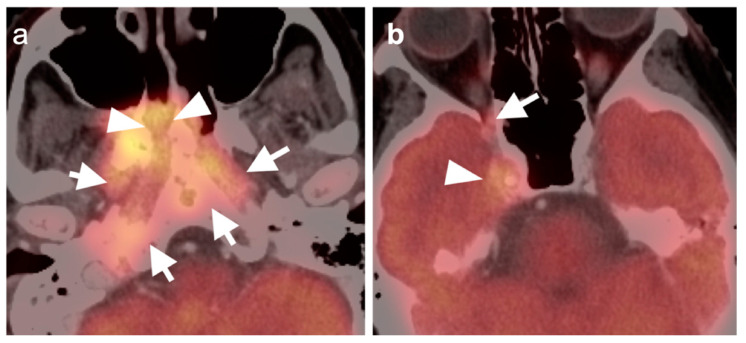
Tumor invading the superior pterygopalatine fossa. (**a**) FDG PET-CT axial image shows hypermetabolic tumor in the pterygopalatine fossa, which is widened (white arrowheads). The tumor also invades the central skull base and nasopharyngeal soft tissues (white arrows). (**b**) The tumor extends intracranially into the cavernous sinus (white arrowhead), around the intracranial carotid artery, and enters the inferior orbital fissure (white arrow).

**Figure 11 cancers-14-02726-f011:**
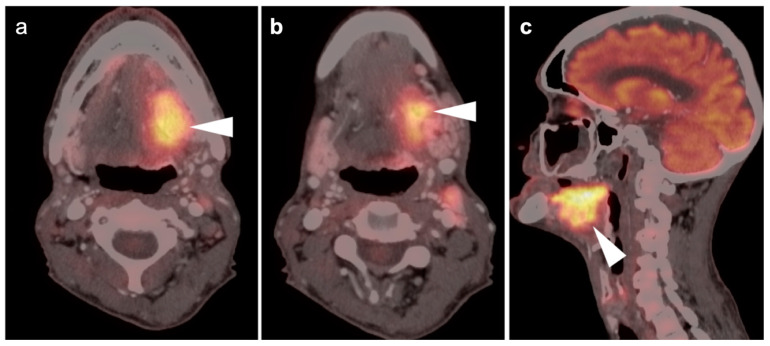
Squamous cell carcinoma of the oral tongue on FDG PET-CT. (**a**) Hypermetabolic tumor involves the left oral tongue (white arrowhead). (**a**,**b**) There is spread to the left floor of mouth musculature shown on both axial (**b**) and sagittal (**c**) imaging (white arrowheads). (**b**) There is also a left level 2A hypermetabolic cervical lymph node consistent with nodal spread of disease.

**Figure 12 cancers-14-02726-f012:**
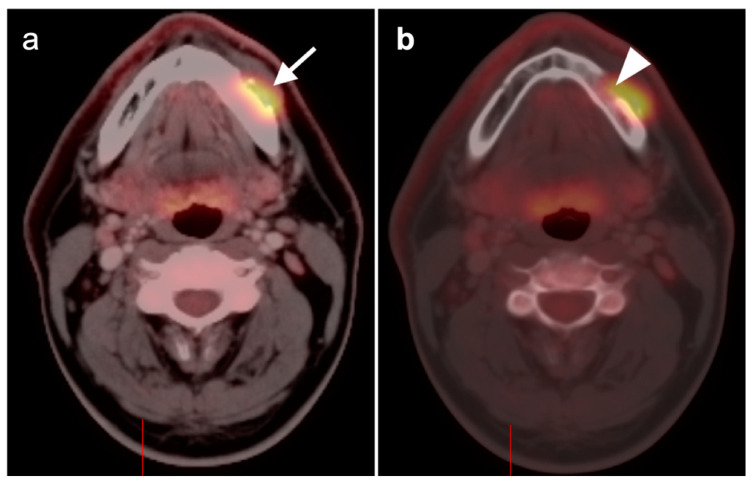
Squamous cell carcinoma of the oral cavity with bony involvement. (**a**) Soft tissue window FDG PET-CT demonstrates a hypermetabolic lesion of the left mandibular alveolar mucosa (white arrow). (**b**) Bone windows demonstrate destruction of the adjacent cortical bone at the mental foramen, consistent with bony involvement (white arrowhead).

**Figure 13 cancers-14-02726-f013:**
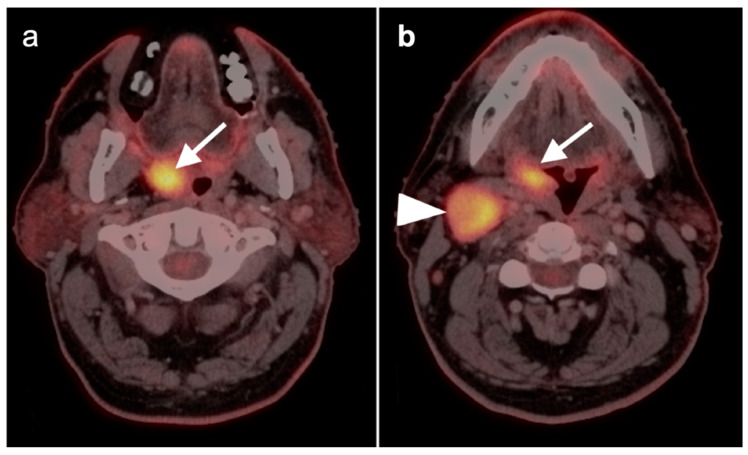
Right tonsillar oropharyngeal squamous cell carcinoma. (**a**) A right palatine tonsillar hypermetabolic squamous cell carcinoma (white arrows). (**b**) There is an enlarged hypermetabolic right level 2 cervical lymph node consistent with tumor involvement ((**b**), white arrowhead).

**Figure 14 cancers-14-02726-f014:**
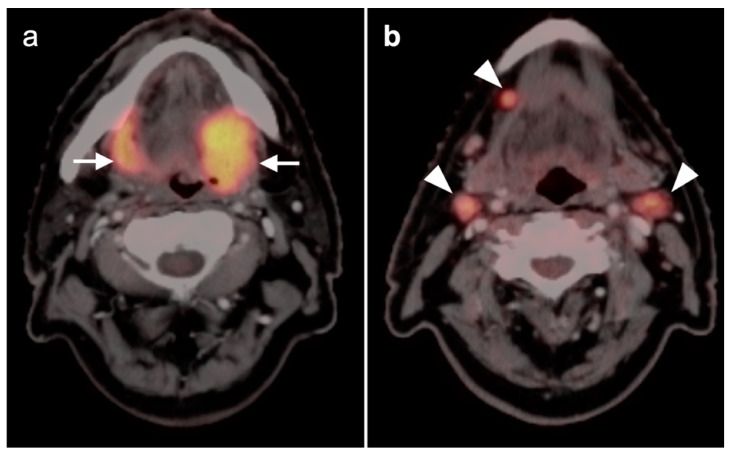
Bilateral oral cavity squamous cell carcinoma of the tongue. (**a**) Right oral tongue squamous cell carcinoma (white arrow) and left glossotonsillar primary with spread into the oral tongue (white arrow); (**b**) There is spread to rounded hypermetabolic right anterior submandibular and bilateral level 2A cervical nodes (white arrowheads).

**Figure 15 cancers-14-02726-f015:**
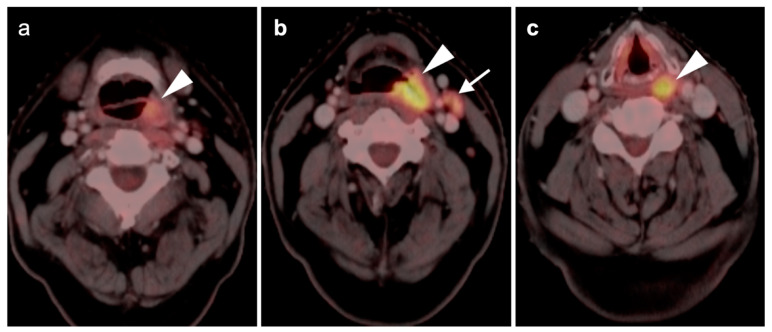
Hypopharyngeal squamous cell carcinoma. (**a**) Hypermetabolic tumor extends superiorly to the left superior aryepiglottic fold (white arrowhead). (**b**) The lesion is centered around the left pyriform sinus (white arrowhead) with involvement of a left level 2A hypermetabolic lymph node (white arrow). (**c**) Inferiorly, the tumor extends to the junction of the hypopharynx and cervical esophagus (white arrowhead) along the lateral wall of the hypopharynx at the level of the cricoid.

**Figure 16 cancers-14-02726-f016:**
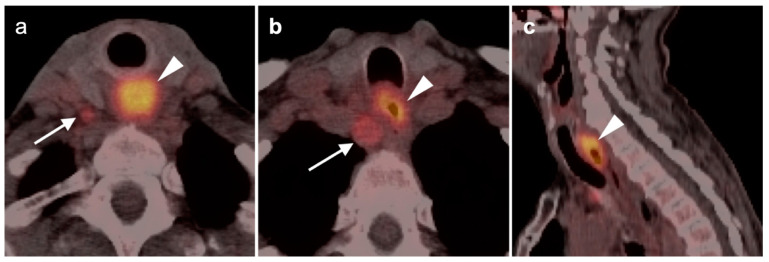
Cervical esophageal squamous cell carcinoma. (**a**,**b**) Axial and (**c**) sagittal FDG PET-CT images demonstrate a hypermetabolic cervical esophageal circumferential mass (white arrowheads) with adjacent hypermetabolic lymph nodes (white arrows).

**Figure 17 cancers-14-02726-f017:**
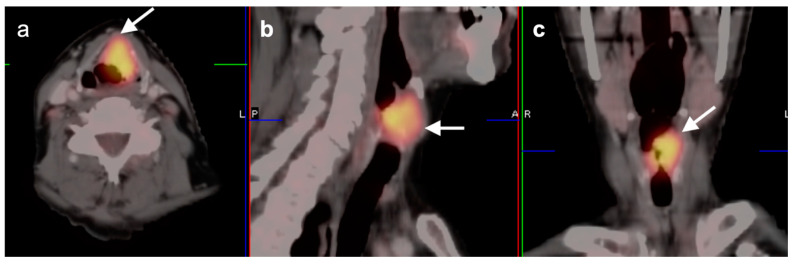
Squamous cell carcinoma of the false cords. (**a**) A large tumor shown in axial (**a**), sagittal (**b**) and coronal (**c**) FDG PET-CT images (white arrows) involves the left false cord and extends across the anterior midline to involve and anterior aspect of the right false cord. The mass effaces the left pyriform sinus and extends to the base of the epiglottis.

**Figure 18 cancers-14-02726-f018:**
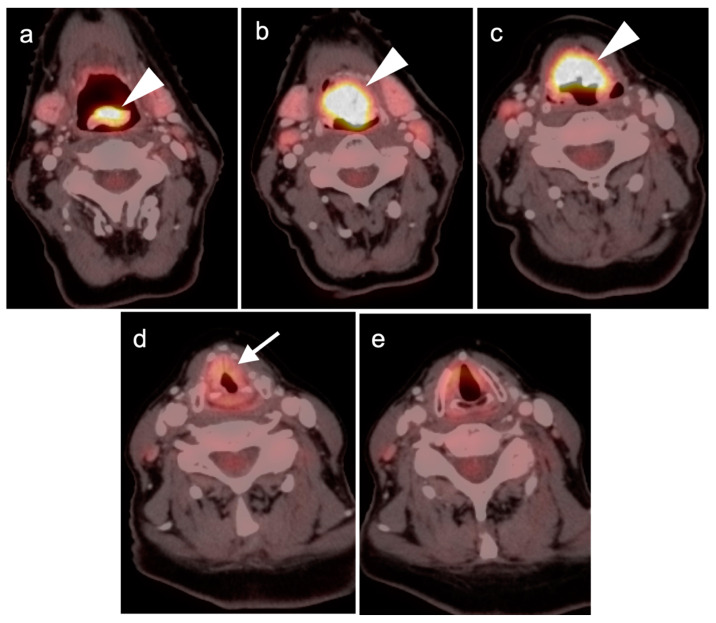
Squamous cell carcinoma of the epiglottis shown on axial FDG PET-CT images. (**a**–**c**) A large hypermetabolic mass involves the entire epiglottis (white arrowheads); (**b**) The tumor fills the preepiglottic space (**c**), white arrowhead); (**d**) There is mild metabolic activity and thickening of the false cords (white arrow) that may be related to either tumor involvement or inflammation/edema; (**e**) The true cords are unremarkable.

**Figure 19 cancers-14-02726-f019:**
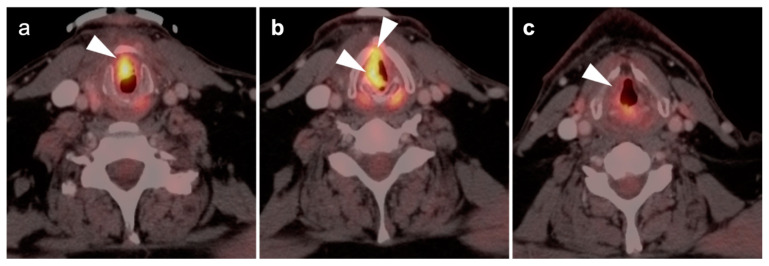
Squamous cell carcinoma of the glottis (true cord). A hypermetabolic tumor of the true cords is shown on axial FDG PET-CT images. (**a**) The lesion involves the anterior commissure (white arrow); (**b**,**c**) The tumor involves the entire right true cord white arrowheads).

**Figure 20 cancers-14-02726-f020:**
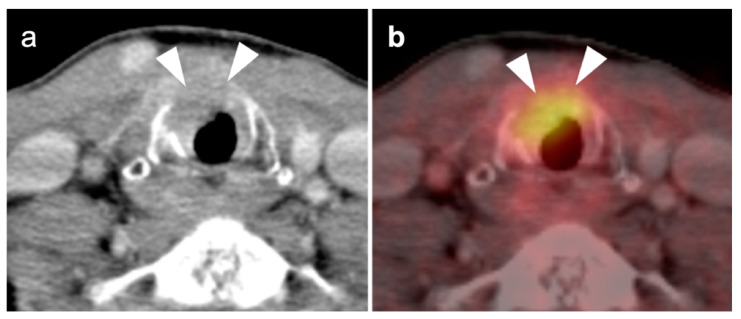
Squamous cell carcinoma of the subglottic larynx. Contrast-enhanced axial CT (**a**) and axial FDG PET-CT (**b**) images show a hypermetabolic soft tissue mass involving the anterior aspect of the subglottic larynx with anterior cricoid destruction (white arrowheads).

**Figure 21 cancers-14-02726-f021:**
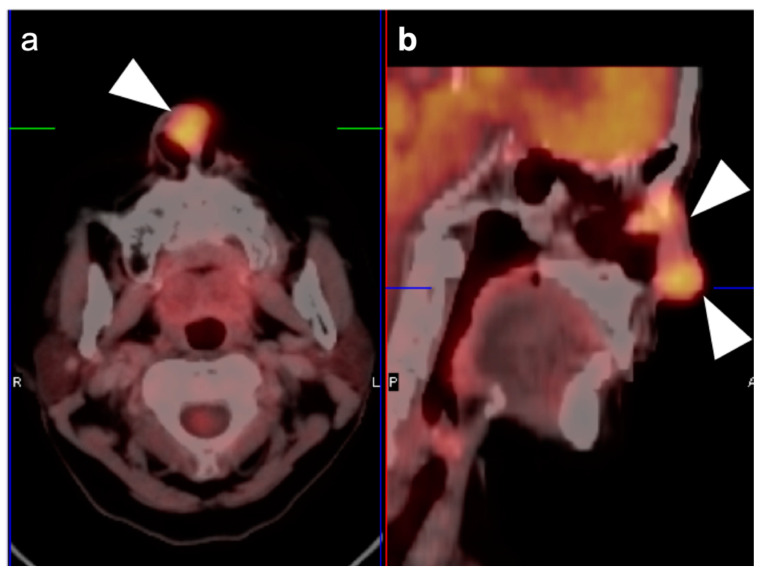
Sinonasal squamous cell carcinoma of the anterior nasal septum. (**a**) Axial FDG PET-CT image demonstrates a hypermetabolic tumor of the anteroinferior nasal septum (white arrowhead). (**b**) Sagittal FDG PET-CT image demonstrates that there is spread throughout the entire nose (white arrowheads).

**Figure 22 cancers-14-02726-f022:**
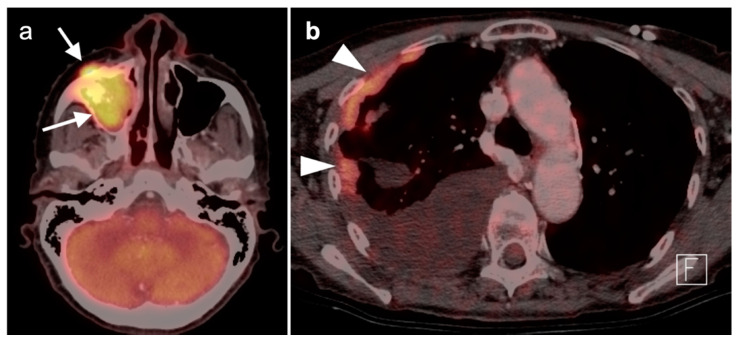
Sinonasal undifferentiated carcinoma (SNUC). (**a**) An intensely hypermetabolic primary tumor in the right maxillary sinus on FDG PET-CT shows extension anteriorly into the pre-maxillary soft tissues (white arrow). (**b**) There are multiple pleural metastases ((**b**), white arrowheads) with a malignant right pleural effusion.

**Figure 23 cancers-14-02726-f023:**
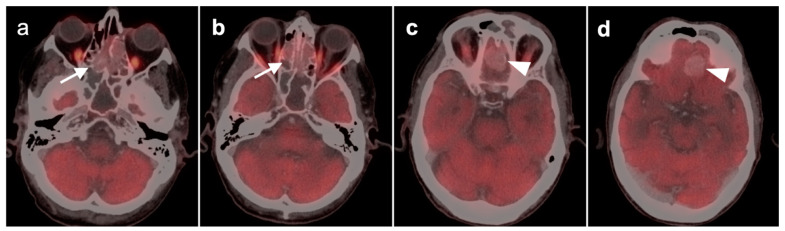
Esthesioneuroblastoma of the ethmoid sinus. (**a**,**b**) Mildly hypermetabolic tumor fills the ethmoid sinuses on axial FDG PET-CT images (white arrows); (**c**,**d**) The tumor extends through the cribriform plate into the anterior cranial fossa/brain (white arrowheads). Without contrast CT, the intracranial extension would be difficult to appreciate on FDG PET.

**Figure 24 cancers-14-02726-f024:**
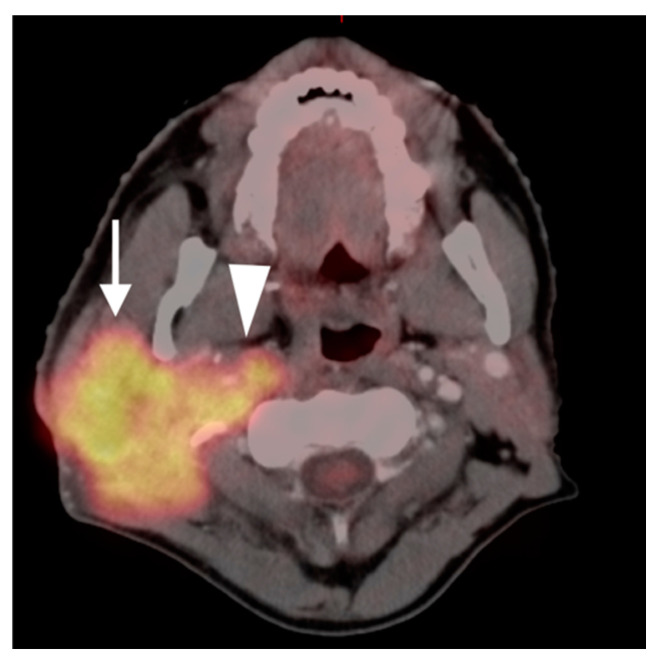
Non-Hodgkin lymphoma of the right parotid gland. On this axial FDG PET-CT image, diffuse large B-cell lymphoma completely fills the superficial (white arrow) and deep (white arrowhead) lobes of the right parotid gland.

**Figure 25 cancers-14-02726-f025:**
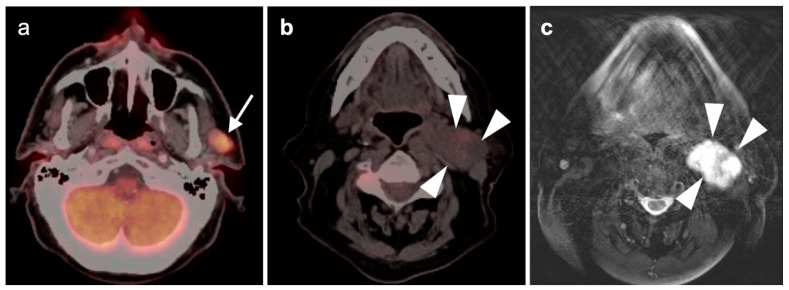
Pleomorphic adenomas, small and large. (**a**) A small left parotid pleomorphic adenoma is strongly hypermetabolic (white arrow). (**b**,**c**) A large pleomorphic adenoma of the left parotid is low in metabolic activity on FDG PET-CT (white arrowheads), (**b**), but is typically bright on T2 MRI (white arrowheads), (**c**).

**Figure 26 cancers-14-02726-f026:**
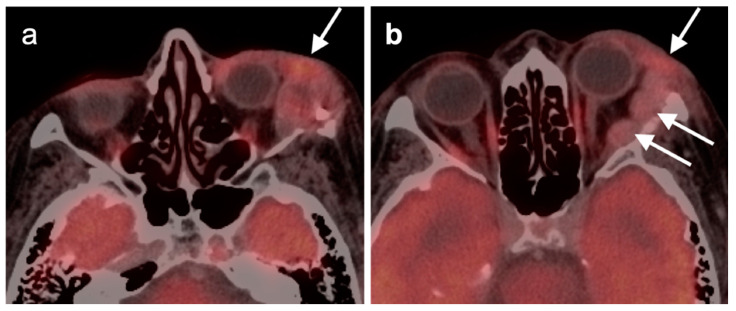
Carcinoma ex-pleomorphic adenoma (CXPA) of the lacrimal gland. (**a**) CXPA of the left lacrimal gland is mildly hypermetabolic (white arrows). (**b**) Tumor extends intraorbitally along the lateral orbital wall with medial displacement of the orbital contents.

**Figure 27 cancers-14-02726-f027:**
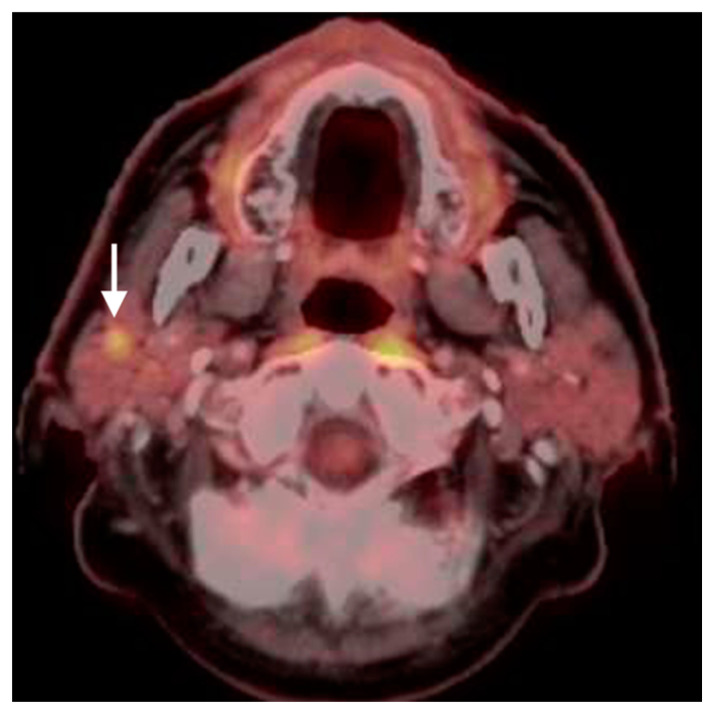
Warthin tumor (papillary cystadenoma lymphomatosum). On axial FDG PET_CT, a Warthin tumor of the right parotid is a hypermetabolic nodule (white arrow).

**Figure 28 cancers-14-02726-f028:**
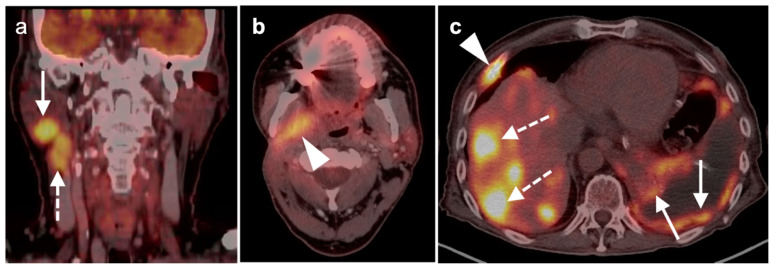
Mucoepidermoid carcinoma (MEC). (**a**) Initial coronal FDG PET-CT image shows a hypermetabolic tumor in the right parotid gland (white arrow) with spread to an adjacent cervical lymph node (white dashed arrow). (**b**) Seven years after initial right neck dissection, there is recurrence in the deep lobe of the right parotid (white arrowhead). (**c**) Also 7 years after initial treatment, there are widespread systemic metastases to the bones (white arrowhead), liver (white dashed arrow) and left pleura (white arrow).

**Figure 29 cancers-14-02726-f029:**
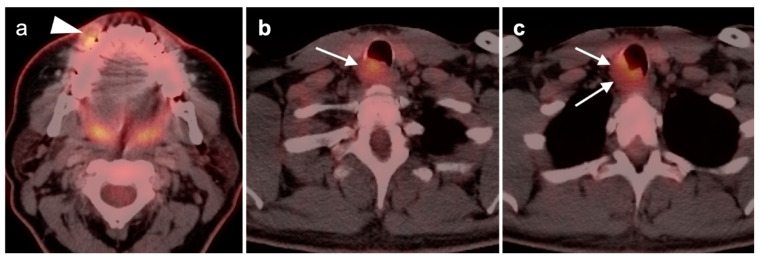
Adenoid cystic carcinomas (ACC). (**a**) ACC of the buccal surface is mildly hypermetabolic on FDG PET-CT (white arrowhead). (**b**,**c**) ACC of the trachea is also relatively mild in metabolic activity (white arrows).

**Figure 30 cancers-14-02726-f030:**
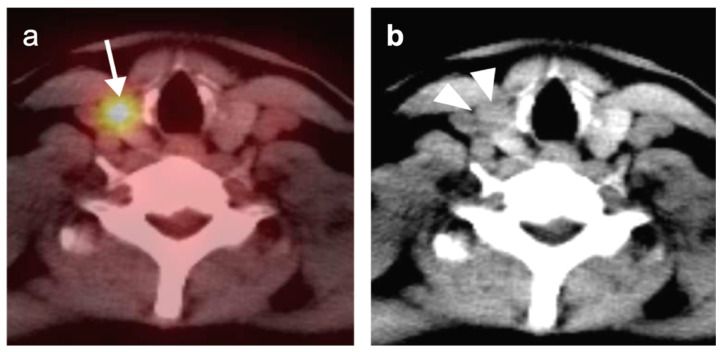
Hypermetabolic thyroid nodule (incidentaloma). A hypermetabolic 1.4 cm nodule on FDG PET-CT ((**a**), white arrow) is lower in attenuation than surrounding thyroid tissue on CT ((**b**), white arrowheads). The American Thyroid Association recommends FNA for hypermetabolic thyroid nodules > 1 cm in diameter.

**Figure 31 cancers-14-02726-f031:**
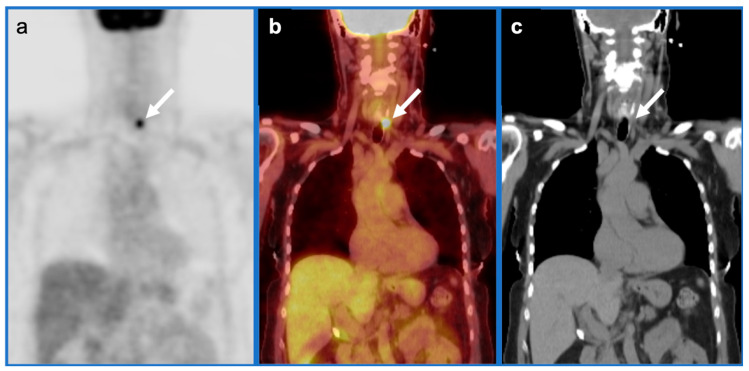
Recurrent differentiated thyroid cancer. Coronal (**a**) FDG PET, (**b**) fused FDG PET-CT, and (**c**) CT demonstrate a small hypermetabolic site of recurrent thyroid cancer along the left aspect of the trachea (white arrows) in a patient with a rising thyroglobulin following surgery and ^131^I NaI ablation for well-differentiated papillary thyroid cancer.

**Figure 32 cancers-14-02726-f032:**
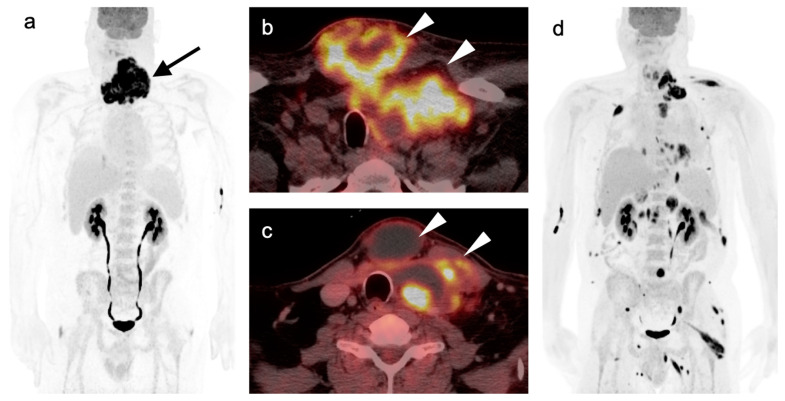
Anaplastic thyroid cancer. (**a**,**b**) At initial diagnosis, tumor is localized to the lower neck, as shown by (**a**) FDG PET MIP (black arrow) and (**b**) FDG PET-CT axial image (white arrow); 6 months later (**c**,**d**), after treatment for anaplastic thyroid cancer, the primary tumor has become smaller and partially cystic ((**c**), white arrows) but portions remain metabolically active. However, there is now widespread metastatic disease as shown by FDG PET MIP (**d**).

**Figure 33 cancers-14-02726-f033:**
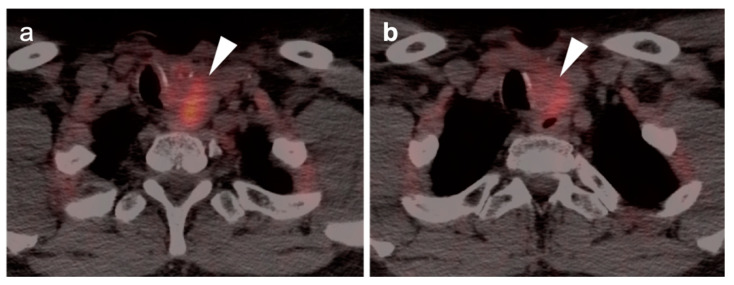
Recurrent medullary thyroid cancer. (**a**,**b**) FDG PET-CT axial images demonstrate a mild-moderately hypermetabolic mass (white arrowheads) in the left thyroid bed in a patient with a rising serum calcitonin level 5 years following total thyroidectomy for medullary thyroid cancer.

**Figure 34 cancers-14-02726-f034:**
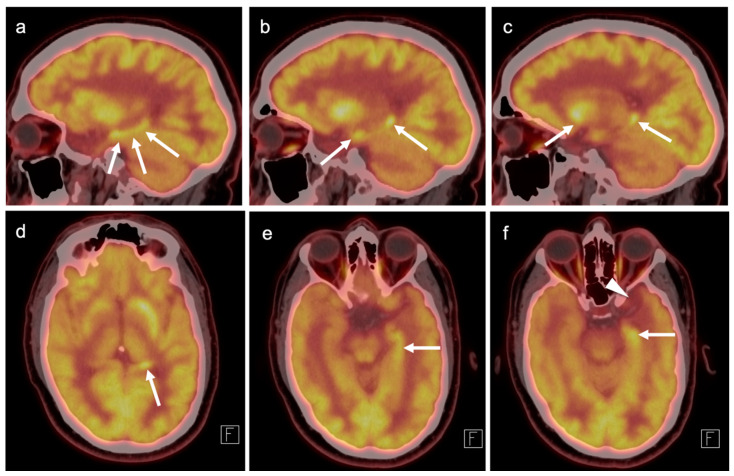
Limbic encephalitis. Patient presented with diplopia, dizziness, headaches, behavioral and memory disturbances. (**a**–**c**) Sagittal FDG PET-CT images show increased metabolic throughout the entirety of the hippocampus (white arrows). (**d**–**f**) Axial FDG PET-CT images show increased metabolic activity throughout the entire left hippocampus (white arrows). There is also decreased metabolic activity in the anterior left temporal lobe (white arrowhead). Limbic encephalitis can either be autoimmune or paraneoplastic in etiology.

**Figure 35 cancers-14-02726-f035:**
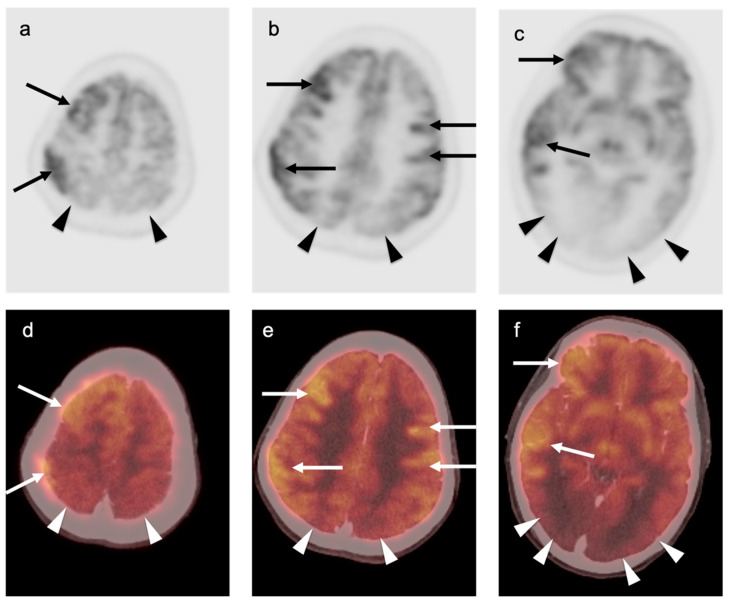
Anti-NMDAR encephalitis. The patient presented with seizures and psychiatric disturbances. Anti-NDMAR Ab titers: serum 1:40, CSF 1:120. MRI imaging was unremarkable. Whole-body FDG PET-CT imaging was negative for sites of tumor involvement, so that the etiology was likely autoimmune. (**a**–**c**) Axial FDG PET images demonstrate scattered cortical areas of increased metabolic activity (black arrows) with global decreased metabolic activity within the posterior cerebrum (black arrowheads). (**d**–**f**) Fused axial FDG PET-CT images demonstrate similar findings to non-fused imaging, with scattered cortical areas of hypermetabolism (white arrows) and global decreased metabolic activity in the posterior cerebrum (white arrowheads). Anti-NMDAR encephalitis can be either paraneoplastic or autoimmune in etiology.

**Figure 36 cancers-14-02726-f036:**
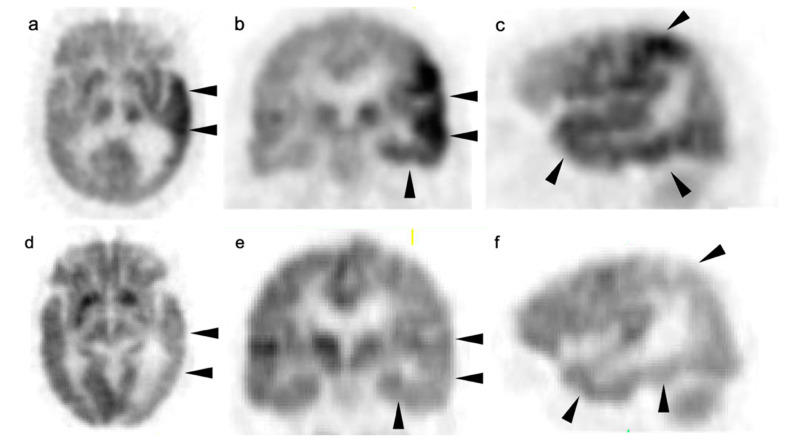
Mental status changes in a patient being treated for an underlying malignancy, proving to be herpes simplex encephalitis (HSE) at biopsy. (**a**–**c**) Increased metabolic activity in the left temporal lobe is typical for acute phase of HSE (black arrowheads). (**d**–**f**) Repeat PET-CT 6 months following recovery of HSE shows resultant decreased metabolic activity due to neuronal damage (black arrowheads). Viral encephalitis can mimic either paraneoplastic encephalitis or primary brain tumor on FDG PET.

**Figure 37 cancers-14-02726-f037:**
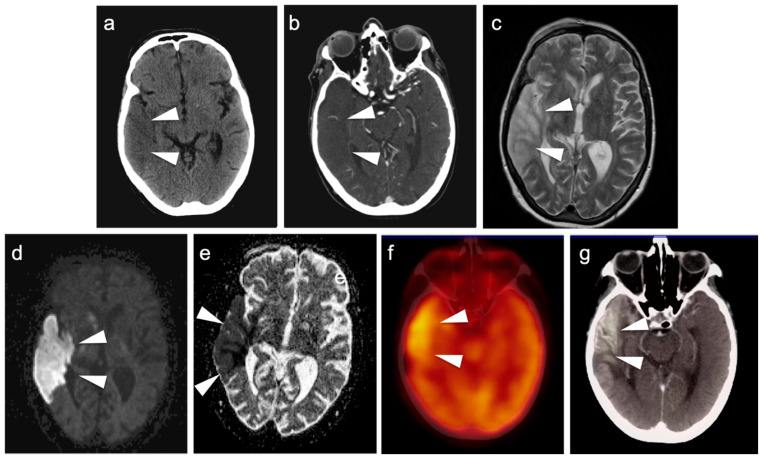
Subacute stroke with subacute increased uptake of FDG may mimic brain tumor. (**a**) Acute stroke with hypoattenuation in the right temporal lobe on con-contrast-enhanced CT (white arrowheads). (**b**) Acute stroke on contrast-enhanced CT with decreased vascularity in area of right temporal lobe infarction (white arrowheads). (**c**) T2 MRI in acute stroke showing increased signal in right temporal lobe due to edema (white arrowheads). (**d**) Diffusion weighted MRI (DWI) image shows increased signal in area of the right temporal stroke in the same patient (white arrowheads). (**e**) Apparent diffusion coefficient (ADC) MRI map shows a corresponding area of decreased signal in the right temporal lobe in the same patient (white arrowheads). (**f**,**g**) FDG PET-CT scan in a subacute stroke (2 weeks after presentation, performed because of co-existing vulvar carcinoma) shows increased metabolic activity coinciding with the region of CT high-attenuation pseudolaminar necrosis in the right temporal lobe (white arrowheads).

**Figure 38 cancers-14-02726-f038:**
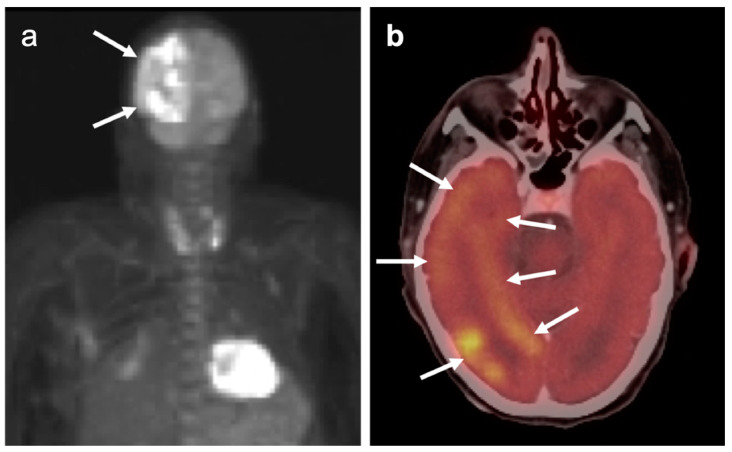
Active seizure on FDG PET-CT in a patient with paraneoplastic hypercalcemia. (**a**) The MIP image of an FDG PET-CT demonstrates right hemispheric hypermetabolism (white arrows) in a patient with an active seizure (the patient was sedated for the exam). (**b**) An axial PET-CT image demonstrates right hemispheric heterogeneous activity due to the seizure activity (white arrows). Active seizure activity may mimic a tumor, and MRI and conventional imaging are critical in making the correct diagnosis.

**Figure 39 cancers-14-02726-f039:**
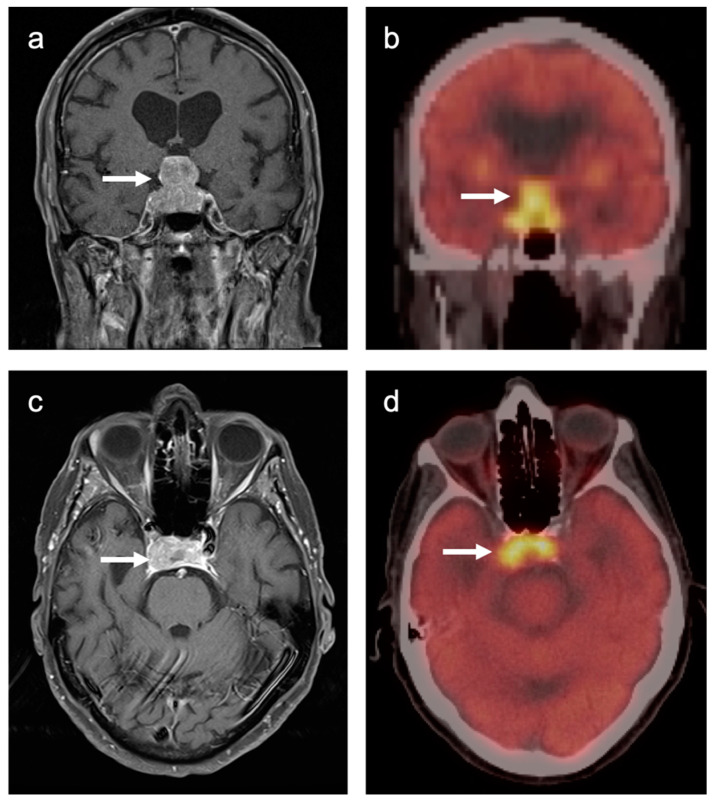
Pituitary macroadenoma. (**a**) Coronal Gd+ T1 MRI and (**b**) coronal FDG PET-CT images demonstrate a sellar-suprasellar hypermetabolic and Gd-enhancing mass (white arrow), unchanged over several years. (**c**) Axial Gd+ T1 MRI and (**d**) axial FDG PET-CT images demonstrate a sellar-suprasellar hypermetabolic and Gd-enhancing mass (white arrow). The patient had a corresponding stable subtle bitemporal hemianopia due to optic chiasm mass effect.

**Figure 40 cancers-14-02726-f040:**
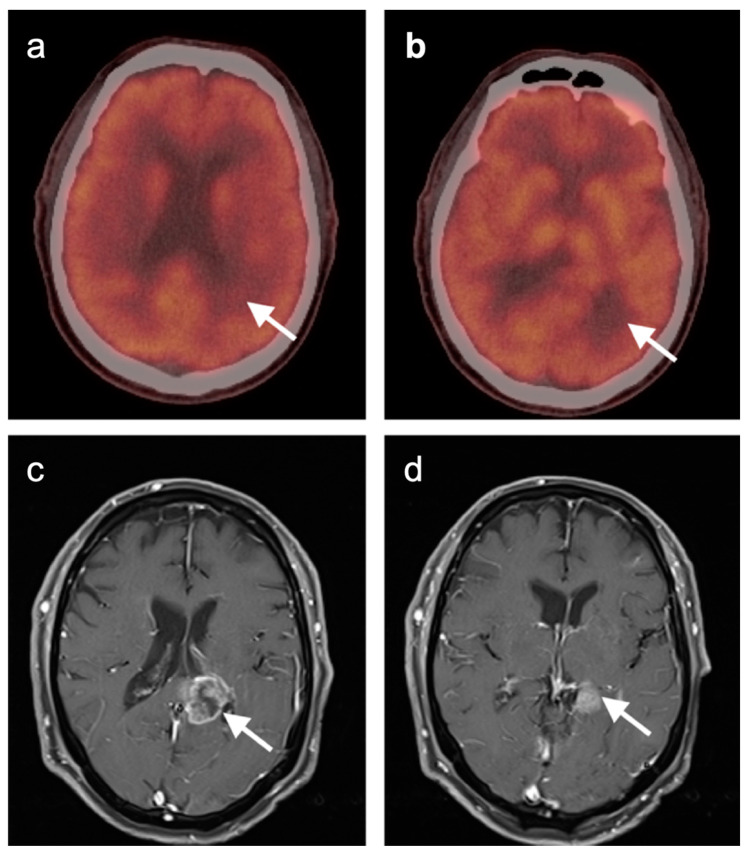
High-grade glioma with low FDG uptake. (**a**,**b**) FDG PET-CT scan shows very low uptake in the enhancing mass, which was a high-grade glioma by biopsy. (**c**,**d**) Gd-enhanced T1 fat saturation axial MRI shows an enhancing mass adjacent to the posterior horn of the left lateral ventricle (white arrows). The patient was on high-dose steroids, although it was not clear whether this was the cause of the low metabolic activity in the tumor.

**Figure 41 cancers-14-02726-f041:**
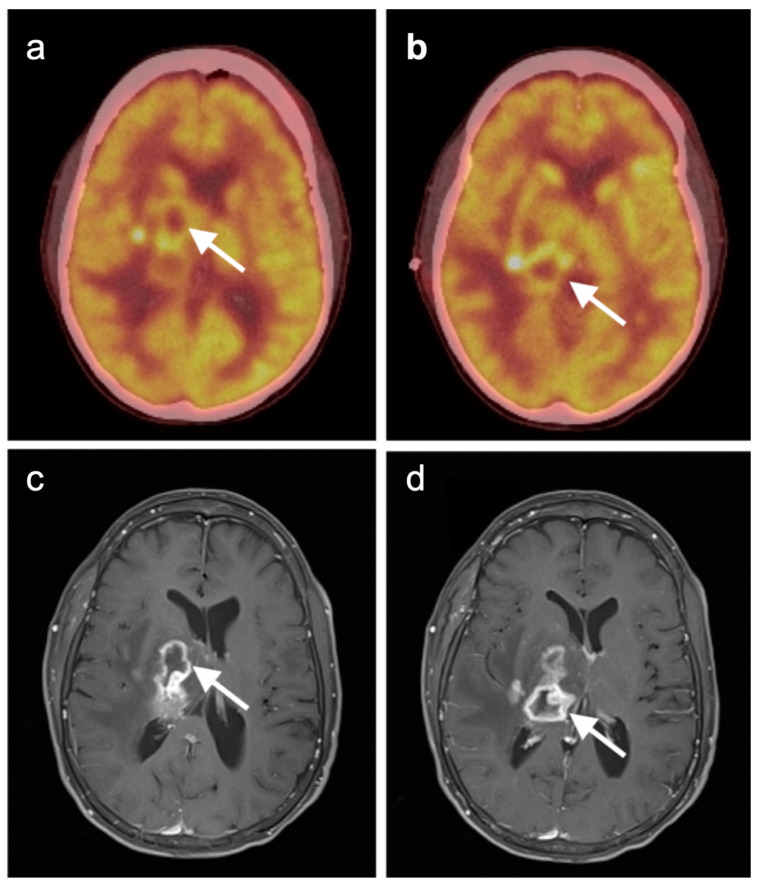
Diffuse large B-cell lymphoma. (**a**–**d**) Findings on MRI in an immunocompromised patient raise the question of infection (such as toxoplasmosis) versus CNS tumor, particularly lymphoma. (**a**,**b**) Multiple lesions are peripherally hypermetabolic on axial FDG PET-CT (white arrows) and were biopsy-proven to be DLBCL (**c**,**d**) Gd-enhanced fat saturation axial T1 MRI images show multiple ring-enhancing lesions (white arrows).

**Figure 42 cancers-14-02726-f042:**
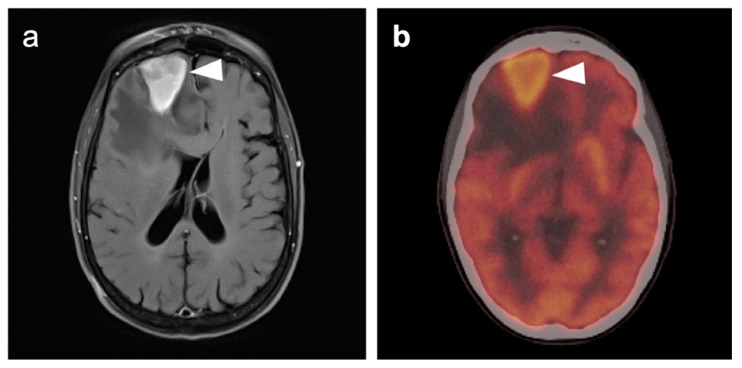
Primary CNS lymphoma. (**a**) Enhancing lesion in the right frontal lobe on axial contrast-enhanced Gad+ T1FS MRI, with surrounding edema and mass effect (arrowhead); (**b**) The lesion shows intense activity on axial fused FDG PET-CT (arrowhead). This was a biopsy-proven primary CNS lymphoma.

**Figure 43 cancers-14-02726-f043:**
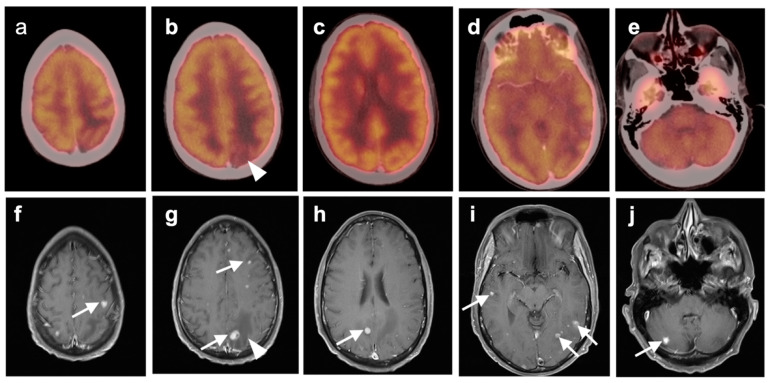
Brain metastases, isometabolic to normal brain cortex (melanoma). (**a**–**e**) Incremental axial FDG PET-CT images demonstrate some hypometabolism in areas of edema (white arrowhead) but multiple melanoma metastases to the brain are not resolved as having metabolic activity greater than that of normal brain parenchyma; (**f**–**j**) Incremental axial Gd-enhanced T1 MRI images demonstrate multiple enhancing brain nodules which were due to metastatic melanoma (white arrows).

**Figure 44 cancers-14-02726-f044:**
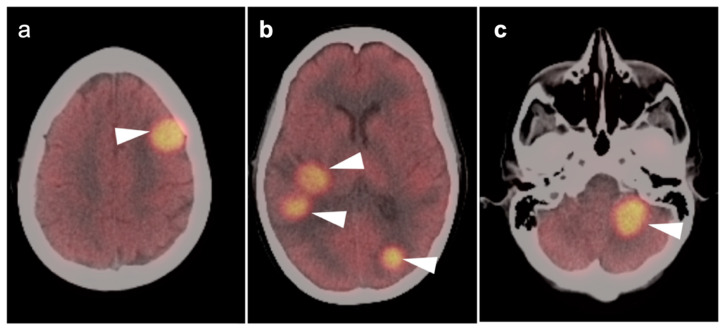
Hypermetabolic brain metastases (breast cancer). (**a**–**c**) There are multiple intensely hypermetabolic nodules in the brain on axial fused FDG PET-CT images (white arrowheads). These were breast cancer metastases.

**Figure 45 cancers-14-02726-f045:**
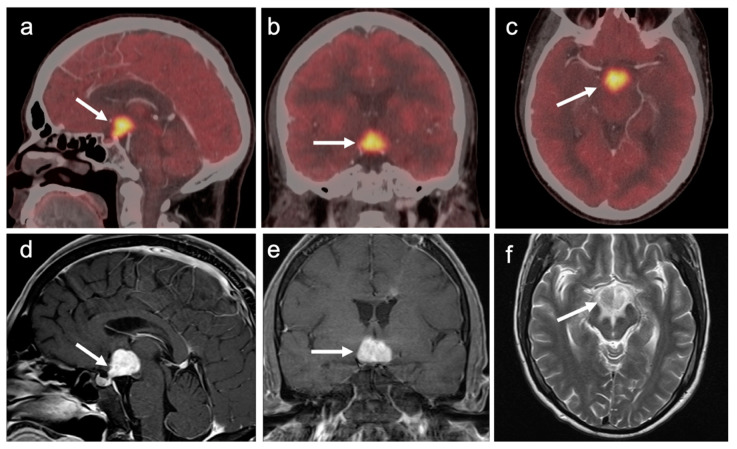
Hypothalamic metastasis from melanoma. (**a**–**c**) Sagittal, coronal and axial FDG PET-CT images demonstrate intensely hypermetabolic hypothalamic mass (white arrows). (**d**,**e**) Sagittal and coronal Gd-enhanced T1 MRI and (**f**) axial T2 MRI demonstrate a corresponding enhancing hypothalamic mass, proven at biopsy to be a melanoma metastasis. Although rare, metastases to the hypothalamic–hypophyseal axis do occur and have been described with melanoma [137,138].

**Figure 46 cancers-14-02726-f046:**
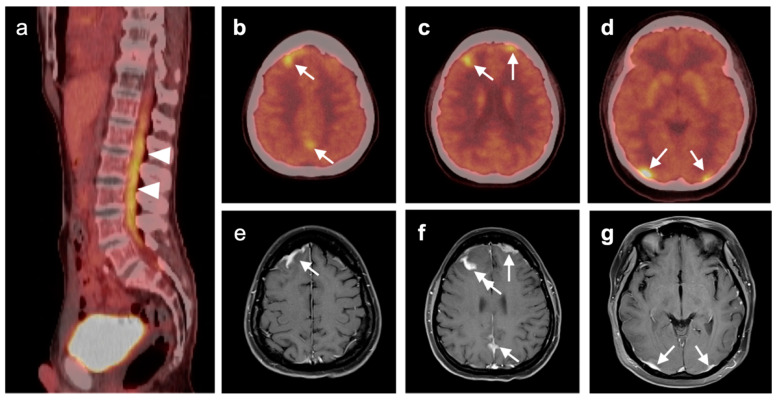
Leptomeningeal carcinomatosis (breast cancer). (**a**) Sagittal FDG PET-CT shows hypermetabolism within the spinal canal due to leptomeningeal disease (white arrowheads); (**b**–**d**) Axial FDG PET-CT images show intense focal areas of increased metabolic activity are noted over the convexities of the brain (white arrows). (**e**–**g**) Axial Gd+ FST1 MRI images show intense enhancement at the sites of increased metabolic activity (white arrows) over the convexities, confirming leptomeningeal carcinomatosis.

**Figure 47 cancers-14-02726-f047:**
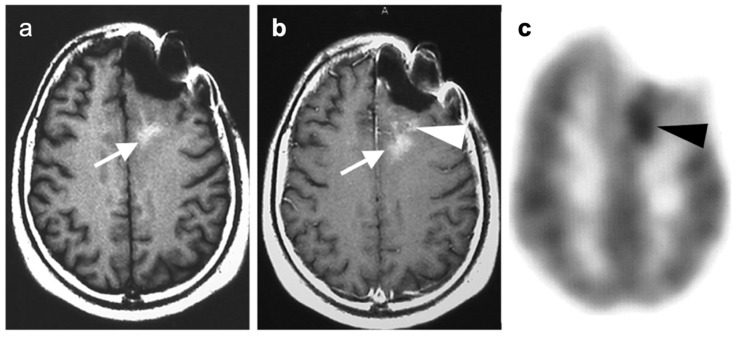
Recurrent high-grade glioma following surgery and radiation. (**a**) T1 non-contrast-enhanced MRI demonstrates a high signal area (white arrow) that does not enhance and is consistent with a region of gliosis. (**b**) Gd enhanced T1 MRI shows enhancement of a nodular region anterior to the region of gliosis, at the margin of the resection cavity (white arrowhead). (**c**) Axial FDG PET image shows that the enhancing region on MRI demonstrates prominent metabolic activity, consistent with recurrent glioma (black arrowhead).

**Figure 48 cancers-14-02726-f048:**
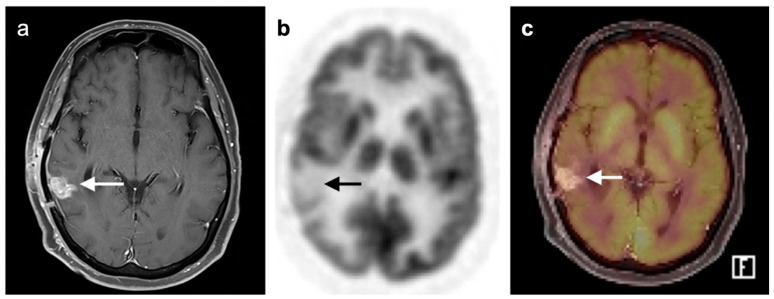
Radiation necrosis. (**a**) Axial Gd enhanced fat suppression T1 MRI shows a region of enhancement in the posterior right temporal lobe in a patient with a high-grade glioma 8 weeks following radiation treatment (white arrow). (**b**) The region shows low activity on axial FDG PET (black arrow), consistent with radiation necrosis. (**c**) Fused axial PET-MRI images help localize the region of interest precisely (white arrow).

**Figure 49 cancers-14-02726-f049:**
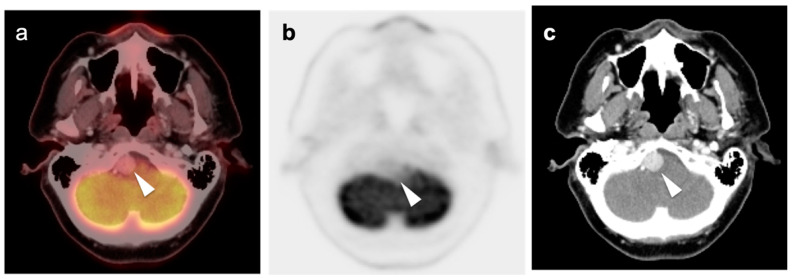
Meningioma. (**a**) Axial fused FDG PET-CT image demonstrates a right posterior fossa hypometabolic enhancing mass with broad-based abutment of the basion (white arrowhead). (**b**) Axial PET only image demonstrates hypometabolism of the mass (white arrowhead), compared to adjacent normal cerebellum, an indicator that this is a lower grade meningioma. (**c**) Axial contrast-enhanced CT confirms the enhancing mass, typical in appearance for a meningioma (white arrowhead).

**Figure 50 cancers-14-02726-f050:**
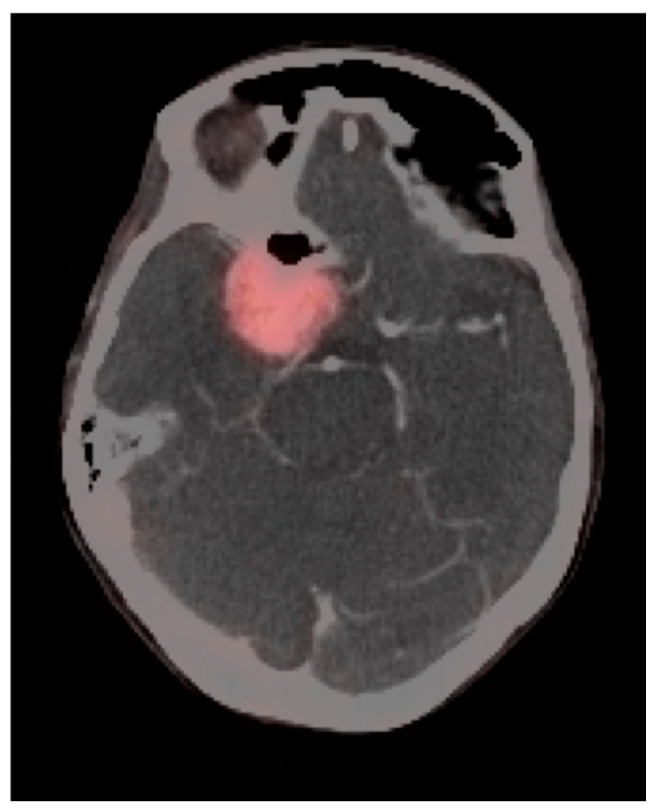
Meningioma on ^68^Ga DOTATATE (NETSPOT^®^) PET-CT. An axial NETSPOT^®^ PET-CT performed with contrast-enhanced CT demonstrates a strongly PET-positive mass in the medial middle cranial fossa, corresponding to a contrast enhancing soft tissue mass with broad-based abutment of the sphenoid bone, typical in appearance for a meningioma.

**Figure 51 cancers-14-02726-f051:**
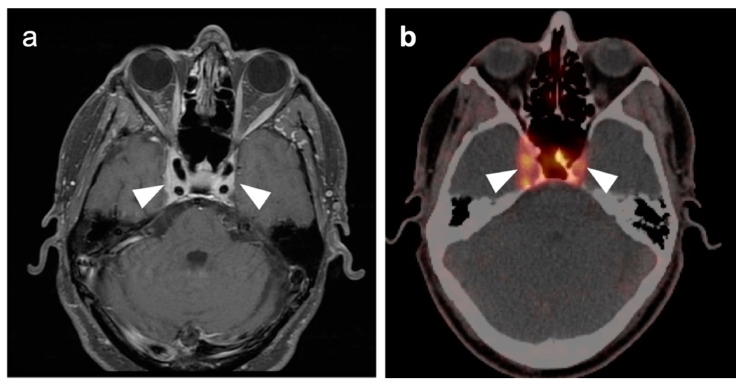
Meningioma on ^64^Cu DOTATATE (Detectnet^®^) PET-CT. (**a**) An axial Gd+ fat saturation T1 MRI image shows bilateral enhancing masses in the cavernous sinuses (white arrowheads). (**b**) ^64^Cu DOTATATE (Detectnet^®^) PET-CT demonstrates concordant increased uptake in the meningioma in the bilateral cavernous sinus (white arrowhead).

**Figure 52 cancers-14-02726-f052:**
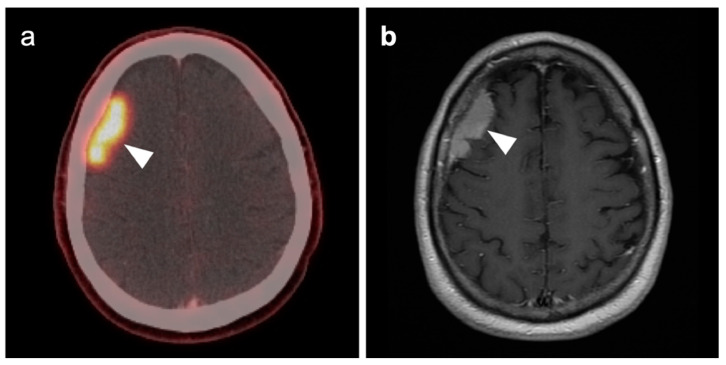
Meningioma on ^18^F fluciclovine (Axumin^®^) PET-CT. (**a**) An Axumin^®^ PET-CT performed for evaluation of biochemically recurrent prostate cancer, incidentally noted to have an intensely PET-positive right frontal dural-based mass (white arrowhead). (**b**) This mass showed homogenous enhancement on Gd+ T1 MRI, arising from the dura, typical in appearance for a meningioma (white arrowhead).

**Table 1 cancers-14-02726-t001:** Hopkins criteria for reporting head and neck cancer response to therapy. Focal uptake in the lesion of interest is compared to that in the internal jugular vein (IJV) and the liver [6].

Hopkins Score	FDG Uptake	Interpretation
1	Focal uptake < IJV	Complete metabolic response
2	Focal uptake > IJV but <liver	Likely complete metabolic response
3	Diffuse uptake > IJV and <liver	Likely post-treatment inflammation
4	Focal uptake > liver	Likely residual tumor
5	Focal uptake markedly > liver	Residual tumor

**Table 2 cancers-14-02726-t002:** NI-RADS criteria for reporting head and neck cancer primary site response to therapy or post-treatment surveillance by FDG PET-CT. Imaging findings, likely clinical significance and suggestions for further management are included [7].

Category	Interpretation	Imaging Findings	Management Recommendation
0	Incomplete	New baseline without prior comparisons available	Compare to previous imaging
1	No evidence of recurrence/persistence	Expected post-treatment changes, no abnormal FDG uptake	Routine surveillance
2a	Low suspicion of recurrence/persistence (endoscopically superficial lesion)	Focal mucosal enhancement, mild to moderate mucosal FDG uptake	Direct visual inspection
2b	Low suspicion of recurrence/persistence (deep lesion)	No discrete mass and either no, mild or moderate FDG uptake	Short imaging follow-up
3	High suspicion of recurrence/persistence	New or enlarged mass or node, enhancement and intense FDG uptake	Biopsy
4	Definite recurrence/persistence	Histologically proven or definite progression	Clinical management

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
