# Peer review of "PET-CT in Clinical Adult Oncology—V. Head and Neck and Neuro Oncology"

_cancers, 2022, doi:10.3390/cancers14112726_

Round 1

Reviewer 1 Report

Positron emission tomography (PET), typically combined with computed tomography (CT) has become a critical advanced imaging technique in oncology. PET-CT has a variety of applications in oncology, including staging, therapeutic response  assessment, restaging and surveillance. FDG PET-CT is an essential modality in the evaluation of head and neck cancer (HNCa). The most successful imaging approach is one that maximizes image quality and ensures that the head and neck are immobilized for the study. The anatomy of the head and neck is complex. Expertise in being able to identify important small structures, and in understanding the regional significance, patterns of spread and distinguishing features of each of the types of HNCa is critical in maximizing the value of FDG PET-CT. Neuro oncologic applications of PET-CT, whether with FDG or some of the newer FDA approved agents, is an ever-evolving field. In most cases, PET-CT is best regarded as an adjunct to conventional imaging by MRI or CT. Maximization of the value of PET-CT in neuro oncology is best met in an interdisciplinary approach between neuro radiologists and nuclear medicine physicians/nuclear radiologists. A clear understanding of the patient history is critical to the correct interpretation of the images. This review article provides an overview of the value, applications, and imaging and interpretive strategies of PET-CT in the more common adult malignancies, including PET-CT imaging application in head and neck neuro oncology.

Author Response

There were no suggestions for revisions provided by this review. Thank you.

Reviewer 2 Report

In this review, the authors have fully discussed the application and limitation of PET-CT in clinical oncology, specifically in Head and Neck cancer (HNCa) and brain tumor. Overall,  they have provided the detailed information of PET-CT as the most successful imaging approach in HNCa and brain tumors.

Author Response

There were no suggestions for revisions provided by this reviewer. Thank you.

Reviewer 3 Report

The first part of the article is a very well written review on FDG PET in Head and Neck tumors, with several pictorial cases. I have only minor comments related to this part.

#1 Last sentence of page 2, beginning with “PET-CT technique…”. The meaning of this sentence is not clear.  

#2 Page 11, Lines 325 and 328; pag 12 line 342.“Bony”: is that grammatically correct?

#3 Page 12 line 337. Please check repetition of “left”.

#4 Page 29 line 863. Check reference 110. It should not be mentioned in the neuro-oncology part as it is on Medullary thyroid cancer.

#5 Pag 30 lines 884, 885. Please check the sentence

The second part on brain tumor is not of the same quality.

Major Comments

#6 First of all, it should be more clearly emphasized that FDG PET is outdated in the evaluation of brain tumors.

#7 The chapters on encephalitis and stroke , with corresponding figures, though illustrative, are not relevant to brain tumors and should be removed.

#8. Page 34, lines 975-977. The authors claim the utility of FDG PET for radiotherapy planning without providing relevant references.  Please note that most updated literature shows that FDG PET is NOT useful for this indication. References:

Galldiks N, et al. Contribution of PET imaging to radiotherapy planning and monitoring in glioma patients - a report of the PET/RANO group. Neuro Oncol. 2021 Jun 1;23(6):881-893

Castellano A, et al. Advanced Imaging Techniques for Radiotherapy Planning of Gliomas. Cancers (Basel). 2021 Mar 3;13(5):1063

#9 Pages 37-38 chapter on brain metastases. A still relevant field of application of FDG in brain tumor is the assessment of treatment-related changes, particularly after radiation treatment of brain metastases. However, this has not been sufficiently highlighted. References:

Chao ST et al.  The sensitivity and specificity of FDG PET in distinguishing recurrent brain tumor from radionecrosis in patients treated with stereotactic radiosurgery. Int J Cancer. 2001;96(3):191–197

Horky LL, et al. Dual phase FDG-PET imaging of brain metastases provides superior assessment of recurrence versus post-treatment necrosis. J Neurooncol. 2011;103(1):137–146.

#10 Same chapter on brain metastases. Please make clear that amino acids are the radiotracers of choice for this indication as well, despite lack of FDG approval.

References:

Galldiks N, et al. PET imaging in patients with brain metastasis-report of the RANO/PET group. Neuro Oncol. 2019;21(5):585–595

Cicone F et al. Long-term metabolic evolution of brain metastases with suspected radiation necrosis following stereotactic radiosurgery: longitudinal assessment by F-DOPA PET. Neuro Oncol. 2021;23(6):1024-1034

Minor comment

#11 Page 35, line 1000. Check reference.
